# Structural details of a Class B GPCR-arrestin complex revealed by genetically encoded crosslinkers in living cells

Yasmin Aydin[1,7], Thore Böttke[1,7], Jordy Homing Lam[2,7], Stefan Ernicke[1], Anna Fortmann[1], Maik Tretbar[3], Barbara Zarzycka[4], Vsevolod V. Gurevich [5], Vsevolod Katritch [2,6] ✉ & Irene Coin [1] ✉

Understanding the molecular basis of arrestin-mediated regulation of GPCRs is critical for deciphering signaling mechanisms and designing functional selectivity. However, structural studies of GPCR-arrestin complexes are hampered by their highly dynamic nature. Here, we dissect the interaction of arrestin-2 (arr2) with the secretin-like parathyroid hormone 1 receptor PTH1R using genetically encoded crosslinking amino acids in live cells. We identify 136 intermolecular proximity points that guide the construction of energy-optimized molecular models for the PTH1R-arr2 complex. Our data reveal flexible receptor elements missing in existing structures, including intracellular loop 3 and the proximal C-tail, and suggest a functional role of a hitherto overlooked positively charged region at the arrestin N-edge. Unbiased MD simulations highlight the stability and dynamic nature of the complex. Our integrative approach yields structural insights into protein-protein complexes in a biologically relevant live-cell environment and provides information inaccessible to classical structural methods, while also revealing the dynamics of the system.

Arrestins are cytosolic proteins that regulate signaling of nearly all G protein-coupled receptors (GPCRs)[1,2]. Besides two "visual arrestins" dedicated to photopigments, two ubiquitous "β-arrestins" (βarr1 and βarr2, a.k.a. arr2 and arr3, respectively) interact in mammals with hundreds of other GPCRs[2]. Arrestins outcompete G proteins at activated and phosphorylated GPCRs, block G protein signaling, and recruit trafficking proteins facilitating receptor internalization[3-10]. Receptor-bound arrestins also initiate several signaling cascades[2,11]. All arrestins share a similar fold consisting of N- and C-domain, each featuring seven β-strands that form a cup-like structure. The central crest on the receptor-binding side is composed of three loops (finger, middle, and C-loop) (Supplementary Fig. 1)[12-15]. GPCRs can form either transient or long-lived complexes with arrestins, which is largely determined by the presence of phosphorylation clusters in their C-terminus[16,17].

Only a few structures of GPCR-arrestin complexes have been solved so far: rhodopsin-arr1 fusion[18-20], and arr2 in complex with neurotensin receptor type 1 (NTS$_1$R)[21,22], chimeric muscarinic acetylcholine receptor M2 (M$_2$R)[23], and beta-1–adrenergic receptor (β$_1$-AR)[24] fused to the highly phosphorylated tail of the vasopressin V2 receptor (V$_2$R-phosphopeptide, V$_2$Rpp). Recently, the structures of the complexes of the full-length V$_2$R[25] and the serotonin receptor 2B[26] with

[1]Faculty of Life Sciences, Institute of Biochemistry, Leipzig University, Bruederstr. 34, 04103 Leipzig, Germany. [2]Department of Quantitative and Computational Biology, University of Southern California, Los Angeles, CA, USA. [3]Medical Faculty, Institute for Drug Discovery, Leipzig University, Bruederstr. 34, 04103 Leipzig, Germany. [4]Division of Medicinal Chemistry, Amsterdam Institute of Molecular and Life Sciences (AIMMS), Faculty of Science, Vrije Universiteit Amsterdam, De Boelelaan 1108, 1081 HZ Amsterdam, The Netherlands. [5]Department of Phar-macology, Vanderbilt University, Nashville, TN 37232-0146, USA. [6]Department of Chemistry, Bridge Institute, USC Michelson Center for Convergent Biosciences, University of Southern California, Los Angeles, CA, USA. [7]These authors contributed equally: Yasmin Aydin, Thore Böttke, Jordy Homing Lam. ✉e-mail: katritch@usc.edu; irene.coin@uni-leipzig.de

arr2 have been published. In all cases, the phosphorylated receptor tail binds to the arrestin N-domain, whereas the 7-transmembrane (7TM) domain holds the arrestin central crest. Notably, the arrestin orientation relative to GPCRs in these structures varies, suggesting that our understanding of arrestin binding to GPCRs is far from comprehensive. Moreover, the structures have major blind spots: the intracellular loop 3 (ICL3) of the GPCR is resolved only in rhodopsin, while the proximal C-tail region downstream of helix VIII is disordered in all cases. No structural data are available for arrestin complexes with receptors belonging to GPCR subfamilies outside of class A (rhodopsin-like).

The parathyroid hormone 1 receptor (PTH1R) is a class B (secretin-like) GPCR activated by endogenous parathyroid hormone (PTH) and by the parathyroid hormone-related protein (PTHrP). It regulates calcium homeostasis, bone development, and turnover[27,28]. PTH1R recruits both β-arrestins forming stable complexes that survive through internalization and even persist in endosomes[29–31]. Structural data are available for agonist-bound PTH1R[32] and for PTH1R bound to the long-acting agonist (PTHLA) and G protein[33]. Downstream of helix VIII (N463–F483), PTH1R features an ~100-residue long flexible C-terminal tail (C-term, K484–M593), which was not resolved in these structures. Several phosphorylation sites have been reported up to residue T551 under different experimental conditions (see Supplementary Table 1). Among these, two phosphorylation clusters, the proximal cluster (S489/S491/S492/S493/S495) and the distal cluster (S501/T503/

S504/T506), as well as a single phosphorylation site at S519 have been consistently reported[34–36].

Here, we characterize the PTH(1-34)-PTH1R-arr2 complex in the natural environment of living cells. By genetically incorporating photo-activatable and electrophilic non-canonical amino acids (ncAAs) throughout the whole arr2, we define the footprint of PTH1R on the arrestin and identify 136 intermolecular pairs of proximal amino acids at the PTH1R-arr2 interface. This large set of spatial constraints is used in extensive conformational sampling to generate optimized all-atom structural models of the PTH1R-arr2 complex, which are stable in unbiased MD simulations. The models reveal unprecedented detail of the PTH1R-arr2 interaction and its dynamic nature.

## Results

### Photo-crosslinking reveals PTH1R footprint on arr2

First, we mapped the footprint of PTH1R on arr2 in live HEK293T cells using the photo-activatable amino acid p-benzoyl-L-phenylalanine (Bpa) as a proximity probe (Fig. 1a)[37–39]. Bpa was incorporated via amber suppression at 416 positions of arr2, from D3 to the last residue R418. Bpa was well tolerated, with 94% of the mutants yielding full-length protein (Supplementary Figs. 2a, 3). Each Bpa-arr2 mutant was co-expressed with wild-type (wt) PTH1R. The receptor was stimulated with PTH(1-34) and the photo-crosslinking was triggered by UV light. When the activated benzophenone moiety comes close to the PTH1R in the receptor-arrestin complex, a covalent bond can form (Fig. 1a).

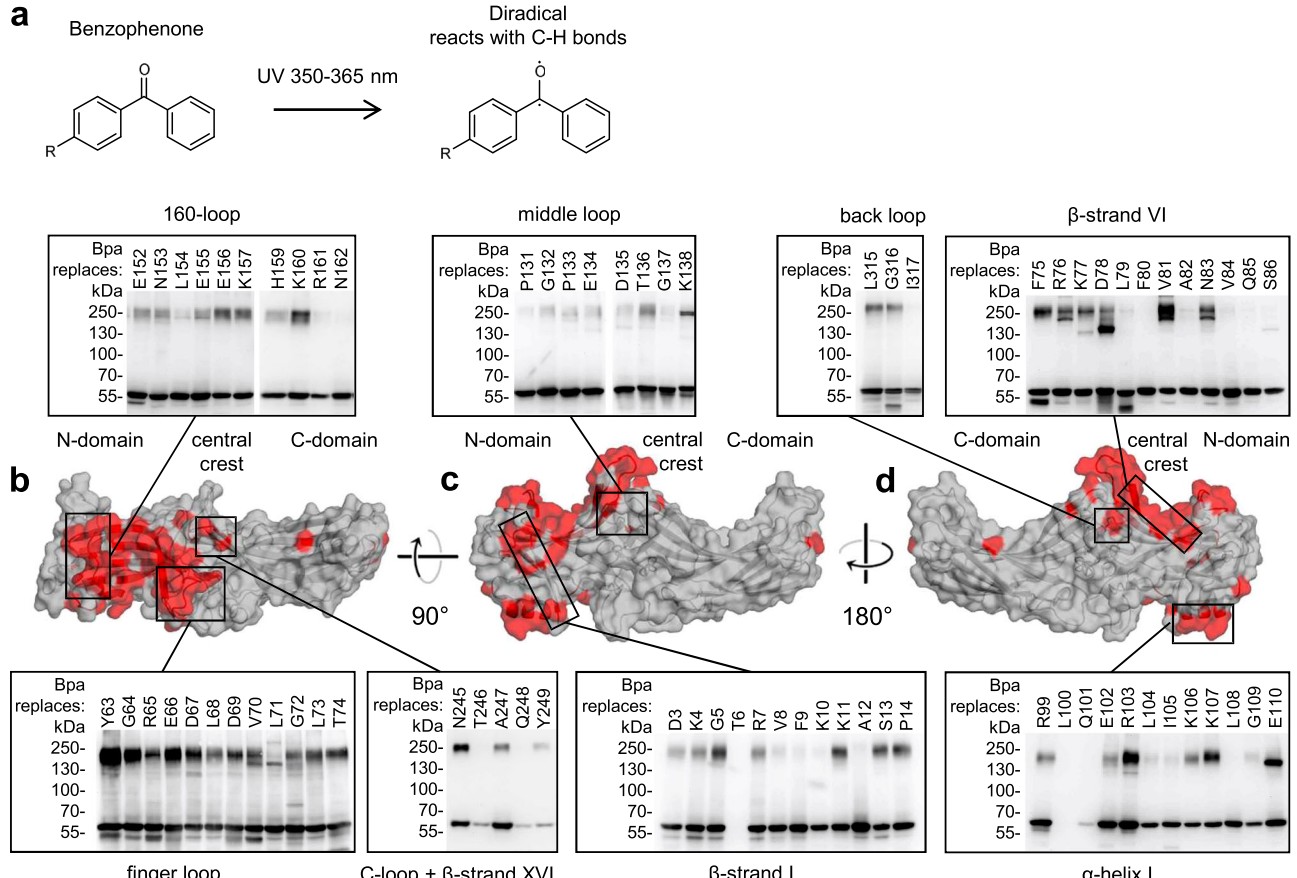

**Fig. 1 | Footprint of PTH1R on arr2. a** Photo-activation of Bpa by UV light. The diradical species inserts into C-H bonds within an estimated radius of ~3.1 Å from the oxygen atom (i.e. up to ~9–10 Å from the Cβ)[37]. **b–d** Surface representation of receptor-bound bovine arr2 (PDBID: 4jqi)[70] with crosslinking hits highlighted in red; the insets are representative western blots of whole cell lysates from photo-crosslinking experiments (*n* = 1) detected with an α-HA antibody (comprehensive overview in Supplementary Fig. 2). Bpa-arr2-3xHA runs at an apparent molecular weight of ~55 kDa, the PTH1R-arr2 complex at ~200 kDa. Top (**b**), front (**c**), and back (**d**) views of arr2 are shown. Bands appearing below 55 kDa are due to intramolecular crosslinking, while other bands at higher MW likely belong to arrestin dimers (band at D78) or possibly other complexes[39].

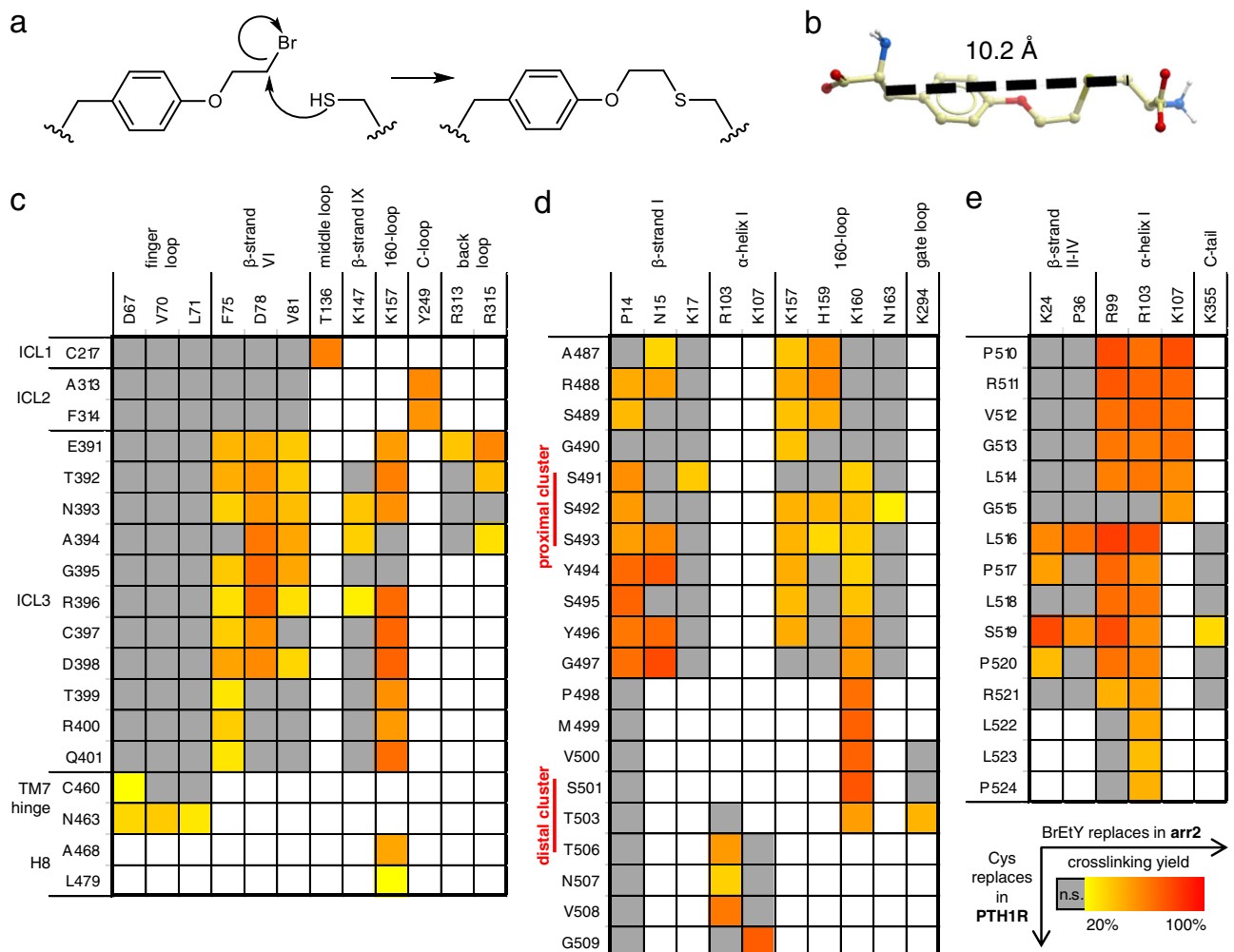

Fig. 2 | Identification of PTH1R-arr2 intermolecular pairs of proximal residues via thiol trapping. a Nucleophilic substitution reaction between the Cys thiol and the haloalkane moiety of BrEtY. The reaction occurs only when the two moieties are close to each other, and yields a stable thioether. b The length of BrEtY crosslink (10.2 Å) was estimated as the maximal distance between Cβ atoms in the BrEtY-cysteine adduct based on the extensive conformational sampling of the adduct molecule. c–e Crosslinking matrix for Cys-PTH1R positions (row) with BrEtY-arr2 positions (column). Fields are colored according to the crosslinking yield of all combinations yielding significant signal over background noise (n ≥ 3, see Supplementary Data 2). Gray squares indicate crosslinking signals not significantly different from control background noise, white indicate combinations that were not tested. All tested combinations are shown in Supplementary Figs. 6, 7.

The covalently linked complex was detected as a ~200 kDa band on western blots, as shown previously[39].

We identified numerous crosslinking positions, most of which were in two regions of arr2, the central crest and the concave side of the N-domain (Fig. 1b–d, Supplementary Fig. 2). In the central crest, major hits were found in the finger loop, in the adjacent β-strand VI (Fig. 1b, d), and in the middle loop (Fig. 1c). In the C-domain near the central crest, hits were detected in the C-loop, β-strand XVI, and the back loop (Fig. 1b, d). Hits in the N-domain were clustered in the 160-loop and β-strand I (Fig. 1b, c), extended up to the distal tip of the convex side of arr2 as well as to α-helix I (Fig. 1d). Two additional hits were detected in the central region of the C-domain (A344) and at the C-edge (N225). The set of arrestin positions photo-crosslinking with PTH1R represents the footprint of the receptor on the arrestin.

**Pairwise crosslinking detects intermolecular proximity points**
Next, we identified intermolecular pairs of proximal PTH1R-arrestin amino acids via pairwise crosslinking. We exploited the reaction between the electrophilic ncAA O-(2-bromoethyl)-tyrosine (BrEtY) and the nucleophilic thiol of canonical cysteine ("thiol trapping" method[40]), which occurs only when the two groups come into close proximity (Fig. 2a)[41].

In the first round, BrEtY was incorporated into 24 positions of arr2 sampling the whole PTH1R footprint (Supplementary Fig. 4a). These BrEtY-arr2 mutants were co-expressed with wt PTH1R, which carries four native Cys residues in the juxtamembrane intracellular elements (Supplementary Fig. 4b). The receptor was stimulated with PTH(1-34), and cells were harvested 90 minutes later. Several BrEtY-arr2 mutants covalently crosslinked to wt PTH1R. Arr2 with BrEtY at six positions located in the finger loop (T58, E66, F75, D78), the middle loop (T136) and the 160-loop (E155) yielded the strongest signals. To identify the captured PTH1R residue, native Cys residues in the receptor were substituted one by one with Ser, which does not react with BrEtY (Supplementary Fig. 4c). The crosslinking signals vanished upon mutating C217 or C397 to Ser, revealing the proximity of E66 and T136 in the finger and middle loops, respectively, to ICL1, and of T58, F75, and D78 in the β-strands surrounding the finger loop, and E155 in the middle loop to ICL3.

In the subsequent rounds, BrEtY-arr2 constructs were combined with a series of PTH1R variants carrying Cys mutations in the intracellular elements. Cys was systematically incorporated throughout the ICL2, the ICL3 (including the intracellular tips of TM5 and TM6), and the whole stretch starting from the intracellular tip of TM7, across helix VIII and the two phosphorylation clusters in the C-tail, up to P553, for a total of 120 positions. BrEtY was incorporated into 51 positions of arr2, starting from

the strongest photo-crosslinking hits and including also positions on the receptor-binding side of the arrestin C-domain. To suppress interference by crosslinking with endogenous Cys in PTH1R, C217 or C397 were substituted by Ser in some blocks. These mutations did not substantially affect receptor function or arrestin recruitment (Supplementary Fig. 5). C150 of arr2 was mutated to Ser to prevent intramolecular crosslinking with BrEtY placed in the N-domain[39]. Substitutions of native Cys residues in arr2 were shown not to affect its receptor binding[42]. The BrEtY-arrestins were combined with Cys-receptors in blocks that were designed based on the topology suggested by the initial search, for a total of 621 combinations. Among these, 443 pairs gave detectable crosslinking signals in western blots (Supplementary Figs. 6 and 7). Experiments were repeated at least three times, western blot signals were quantified by densitometry, and the dataset was analyzed by $t$-test ($p < 0.02$) and Welch-test ($p < 0.05$) against a control set representing the background noise from non-specific binding (Supplementary Fig. 8a, Supplementary Data 1). This yielded 136 crosslinking pairs with signals significantly higher than the control (in most cases > 20% of expressed arr2 crosslinked to PTH1R) (Fig. 2c–e and Supplementary Data 2).

To investigate whether our crosslinking pairs may derive from two distinct populations of complexes either in the "tail" or in the fully engaged "tail + core" conformation, we have incorporated the photo-crosslinker Bpa in the N-domain (loop between β-strands I and II, 160-loop) of an arrestin variant depleted of the finger loop (Y63–K77) (Supplementary Fig. 9), which does not interact with the receptor core[43]. Finger loop-depleted Bpa-arr2 were not able to capture the PTH1R, whereas the corresponding full-length Bpa-arr2 variants yielded strong crosslinking signals in the western blot. These results suggest that the crosslinking constraints reflect predominance of the fully engaged conformation.

The last PTH1R residue involved in a crosslinking pair was P524. To assess whether the distal C-tail interacts with arr2, we generated a set of progressively shortened PTH1R variants and fused them at the C-terminus to NanoLuc (Nluc)[44]. In live HEK293T, we measured BRET signals upon PTH(1-34)–induced recruitment of arr2 N-terminally fused to Venus[45,46]. While the $EC_{50}$ of the PTH1R-arr2 interaction remained stable, the netBRET value (fluorescence/luminescence) constantly increased with the shortening of the receptor C-tail (Supplementary Fig. 10). This is consistent with a reduction of the distance between BRET-donor and -acceptor and suggests that the distal C-tail of PTH1R beyond residue P524 protrudes into the cytosol and does not interact with arrestin.

**Crosslinking-guided modeling of PTHLA-PTH1R-arr2 complex**
The 136 hits from thiol trapping experiments representing pairs of receptor-arrestin positions at an estimated maximal Cβ-Cβ distance of about 10.2 Å (Fig. 2b) were translated into 136 soft harmonic restraints weighted by the corresponding crosslinking yields in an integrative modeling framework[47–49]. The starting 3D models were generated by superimposing the high-resolution structure of the PTHLA-bound PTH1R (G protein-bound active state, PDBID: 6nbf)[33] with structural templates derived from either (1) the rhodopsin-arr1 fusion complex (PDBID: 5w0p)[20], or (2) the M₂R-arr2 complex (PDBID: 6u1n)[23], or (3) the β₁-AR-arr2 (PDBID: 6tko)[24], or (4–5) the two NTS₁R-arr2 complexes (PDBID: 6pwc, 6up7)[21,22]. Regions missing in the templates both of PTH1R (e.g. ICL3 and the C-terminus beyond L481 up to G530) and arr2 were remodeled to their native sequences. The receptor was phosphorylated at S489/S493 in the proximal phosphorylation cluster, at T503/S504 in the distal cluster and at S519, a pattern that is consistently described in the literature (Supplementary Table 1)[34,50,51].

The assembled initial models underwent an extensive energy-based Metropolis Monte Carlo (MMC) sampling in internal coordinates for 10 independent trials each of more than 2 million steps, as described in Methods[47,48,52]. The full MMC sampling involved orientation of the arrestin and 3D conformation of flexible regions including

ICL2 (S308–Y320), ICL3 (L377–F417), helix VIII and the C-terminus (F461–T525) of PTH1R and the N-terminus (M1–R7), crest region (R51–L108, P121–V171, N280–S320) and the C-terminus (P354–K358) of arrestin. All other internal coordinates of the proteins defined by crystal structures (e.g. TM helices of PTH1R and β-strands of arr2) were optimized by fast gradient descent. The conformational sampling converged to M₂R-arr2–like orientations of arrestin for all five templates including even the two NTS₁R-arr2 templates, although in those structures the arrestin engages NTS₁R in an almost perpendicular orientation compared to its pose in the other three complexes (Supplementary Fig. 11a, b). The five models featuring minimal energy of the imposed restraints for each of the templates showed very similar conformations and energies (Supplementary Fig. 11c). The M₂R-arr2–based model gave the best fit to the experimental data with 125 out of 136 crosslinks featuring Cβ-Cβ distances within 15.0 Å (Fig. 3a, Supplementary Data 3, and Supplementary Figs. 12–16). Therefore, it was chosen as the representative static integrative model.

This model predicts several details of PTH1R-arr2 interactions. First, the optimization yielded a minor ~2 Å inward adjustment of TM5/ICL3/TM6 on the intracellular side of the receptor as compared to the G protein-bound PTH1R structure (Supplementary Fig. 17)[53]. Second, a series of ionic and polar interactions were predicted at the PTH1R-arr2 interface (Fig. 4a–e), including E391$^{PTH1R}$-K77$^{arr2}$ (ICL3–β-strand VI, Fig. 4b, Supplementary Fig. 17a), K408$^{PTH1R}$-L71$^{arr2}$ and K405$^{PTH1R}$-G72$^{arr2}$ (TM6-finger loop, Fig. 4c), R219$^{PTH1R}$-E66$^{arr2}$/D69$^{arr2}$ (TM2-finger loop, Fig. 4c), Y249$^{PTH1R}$-F314$^{arr2}$ and N245$^{PTH1R}$-F311$^{arr2}$ (ICL2–C-loop, Fig. 4d). Third, the helix VIII of PTH1R rotated towards the 160-loop of arr2 at the edge of its N-domain, allowing the interaction of pS493$^{PTH1R}$-R161$^{arr2}$ (Supplementary Fig. 17b). This non-canonical conformation is supported by crosslinking of helix VIII residues of PTH1R A468, L479 and proximal positions A468, R467, S489 with K157 and H159 of arr2, by ionic interactions within the receptor helix VIII (Fig. 4e) involving K471$^{PTH1R}$-pS489$^{PTH1R}$, R485$^{PTH1R}$-pS489$^{PTH1R}$/pS493$^{PTH1R}$, as well as by stability of helix VIII in MD simulations (see below). However, our crosslinking dataset cannot exclude an alternative situation, where residues of helix VIII lack secondary structure and are flexible.

The path of the receptor C-terminus is largely defined by the formation of an extended β-sheet between β-strand I of the arrestin (residues V8 to A12) and the β-strand in the PTH1R C-terminus (V500–S504) that overlaps with the distal phosphorylation cluster (S501–T506). The intermolecular β-sheet is stabilized both by backbone interactions and by a network of ionic interactions between the phosphate groups in PTH1R and basic residues of arr2 (Fig. 4a, e.g. pT503-K11, pT503-R25, pT503-K294, and pS504-K10). The proximal PTH1R C-terminus (S489–S495) lacks a secondary structure and shows substantial conformational variations between the models. It is stabilized by interactions at pS493$^{PTH1R}$-R161$^{arr2}$ on the 160-loop (Fig. 4e) in a positively charged region at the edge of the N-domain of arr2, which includes K157, H159, and R165 (160-loop), N15 and K17 (loop between the β-stands I and II) and R51 (loop between β-strands IV and V).

We validated this position of the proximal phosphorylation cluster at the external edge of the N-domain by assessing whether an alternative pose involving interactions with the finger loop, as it was observed in the structure of the arr2-V₂Rpp complex (Supplementary Fig. 18a), is reconcilable with our crosslinking data. First, when crosslinking arr2 with full-length V₂R, we did find strong crosslinking hits in the 160-loop, but not in the finger loop (Supplementary Fig. 18b). Second, when adding distance restraints between the proximal cluster in the PTH1R C-tail and the β-strand VI of arr2 in a modeling experiment (Supplementary Fig. 19a, b), the level of strain in this region of the complex dramatically increased compared to our model (Supplementary Fig. 19c). Overall, these data confirm the path of the proximal phosphorylation cluster of PTH1R C-tail at the N-edge of arr2.

Downstream of the distal cluster, pS519 lies in the proximity of α-helix I of arr2. Although the interactions can switch between several

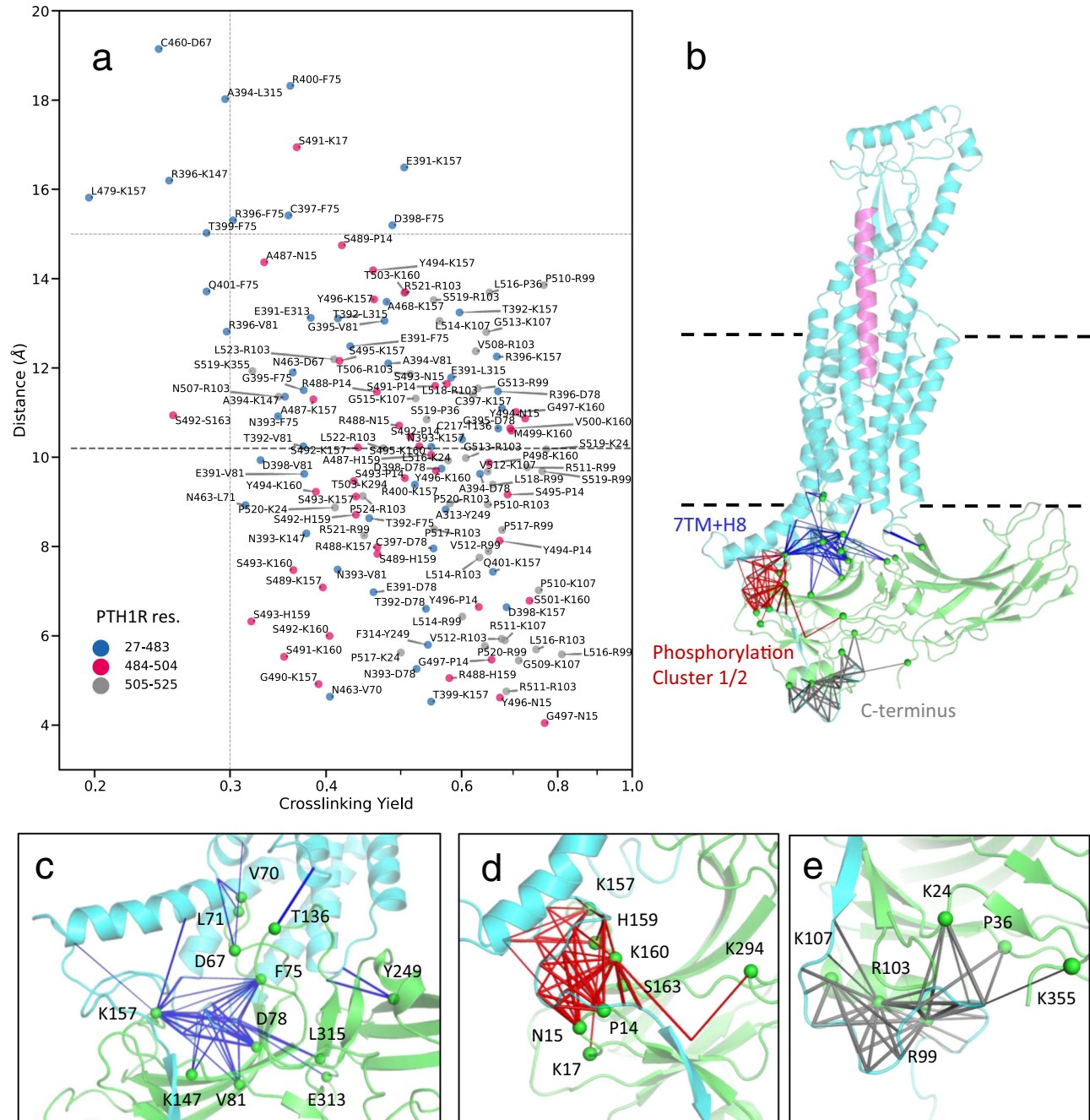

**Fig. 3 | Crosslinking-guided model of PTHLA-PTH1R-arr2 complex. a** Plot of pairwise Cβ-Cβ distances against chemical crosslinking yields in the PTH1R-arr2 model after flexible refinement from the M₂R-arr2 template. For glycine, Cα was used instead of Cβ. The horizontal dashed line at 10.2 Å marks the estimated distance (Cβ-Cβ) for BrEtY-Cys crosslinking, whereas the dotted line at 15.0 Å represents the maximal crosslinking distance when taking into account the flexibility of the complex. The large majority of the crosslinking pairs (125 out of 136), lie within 15.0 Å in the static 3D model, suggesting overall agreement with the crosslinking data. Source data are provided in Supplementary Data 3. **b** Overview of the complex with the location of the crosslinking pairs is shown. Crosslinking pairs in the 7TM and helix VIII of PTH1R (blue, **c**), in the proximal and distal phosphorylation clusters (red, **d**) and in the end of truncated C-terminus (gray, **e**) are shown as connecting solid lines in separate panels; the thickness of each line represents crosslinking yield; residues of arr2 are labeled. For illustrative purposes, a sphere is placed on Cα of labeled arr2 residues instead of the Cβ where the distances are measured.

---

positively charged side-chains of arr2 in the model, this position well satisfies other crosslinking proximity restraints and might contribute to the overall complex stability (Fig. 2).

**Unbiased molecular dynamics suggest stability of the complex**
Because each crosslinking pair is identified in an independent experiment, the proximity requirements for the formation of Cys-BrEtY adduct are not expected to be all satisfied simultaneously by a single conformational snapshot of the flexible complex. For example, some arr2 residues, such as F75, K107, K157, and K160, are crosslinking hubs involved in as many as 8–13 crosslinks, resulting in steric "over-crowding" in static models (Supplementary Fig. 20). While nearly all crosslinking pairs lie within a Cβ-Cβ distance of 15 Å in our model (Fig. 3a and Supplementary Data 3), only 72 out of 139 pairs (~52%) fall within the strict 10.2 Å cutoff, which is the Cβ-Cβ distance in the BrEtY-Cys adduct (Fig. 2b).

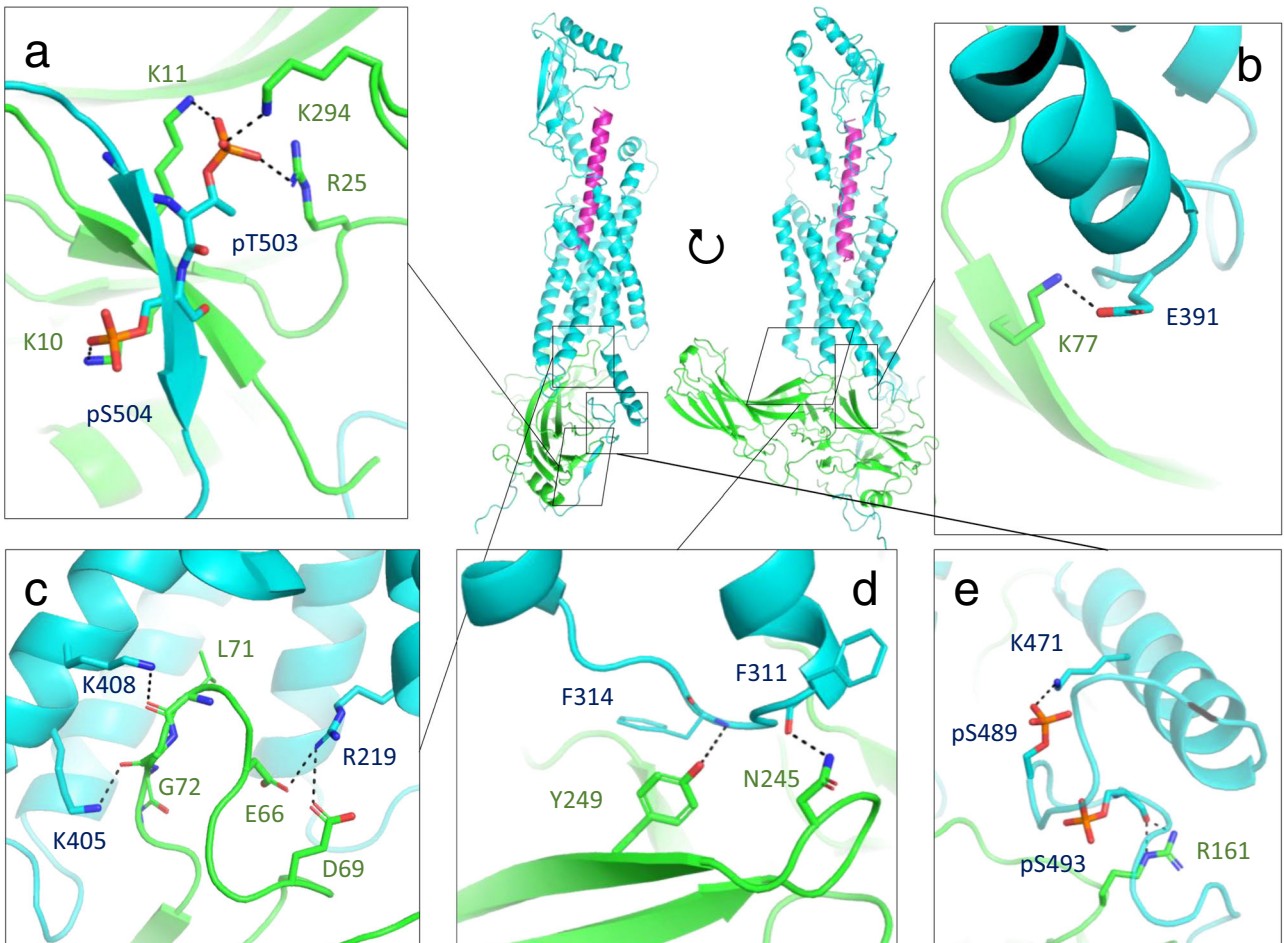

**Fig. 4 | Molecular Interactions in the optimized PTH1R-arr2 model based on the M₂R-arr2 template.** Two views of the model containing long-acting PTH (PTHLA, magenta), arr2 (green), and PTH1R (cyan) are presented at the center. Interactions involving the (**a**) distal phosphorylation cluster, (**b**) ICL3 of PTH1R, (**c**) finger loop, (**d**) C-loop of arr2, and (**e**) proximal phosphorylation cluster of PTH1R are shown in separate panels.

To analyze the dynamic accessibility of the residue pairs for crosslinking, as well as the overall conformational stability of the complex, we performed unbiased Molecular Dynamics (MD) simulations without crosslinking restraints. Ten independent MD trajectories, 1.2 μs each, were obtained for the PTH1R-arr2 model in a lipid bilayer. The vast majority (130 out of 136) of the crosslinked pairs came within the Cβ-Cβ cutoff <10.2 at some point during the MD simulations, including most of the pairs that were >15 Å apart in the static model (Fig. 5a–c, Supplementary Data 4–6). The remaining six outliers were all involved in the crosslinking hubs and still came closer than 15 Å, which is consistent with a margin of about 3–6 Å over the physical crosslinking distance observed in a systematic study of chemical crosslinks[54].

In general, the PTH1R-arr2 coupling was maintained robustly in MD simulations, and interactions were consistent with those predicted by our static snapshot (Fig. 4). However, some interactions at solvent/membrane-exposed region were more dynamic (Supplementary Figs. 21–26). For example, E391$^{PTH1R}$-K77$^{arr2}$ (ICL3-finger loop) was maintained at 62.6% of the cumulated simulation time (Supplementary Fig. 22b). Interestingly, another salt bridge involving R219 in TM2 of PTH1R and the finger loop of arr2 dynamically switched to either E66$^{arr2}$ or D69$^{arr2}$ (Supplementary Fig. 24a). In the phosphorylation clusters, interactions of both the backbone and the phosphate groups remained stable throughout the simulation (Supplementary Figs. 22–23), with the proximal cluster showing higher mobility than the distal cluster (Supplementary Fig. 27). The position of helix VIII was supported by

several intramolecular ionic interaction with the proximal cluster at K471$^{PTH1R}$-pS489$^{PTH1R}$ and R485$^{PTH1R}$-pS489$^{PTH1R}$/pS493$^{PTH1R}$ (Supplementary Figs. 26, 28, 29). The receptor C-terminus downstream of the distal cluster (res. V505–T525) shows higher mobility in MD simulations. Frequent salt bridge interactions were observed for pS519$^{PTH1R}$ with R99$^{arr2}$ and occasional with R103$^{arr2}$, K4$^{arr2}$, and K24$^{arr2}$, suggesting a potential functional role of this phosphorylation site (Supplementary Figs. 25, 30).

We also measured the orientation of arr2 relative to PTH1R (see "Methods" for details). Figure 5d–f shows a modest fluctuation of 13–25 degrees in the orientation during MD simulations (Supplementary Data 7), with the pitch variations being the most limited by the membrane anchoring of arr2 C-edge (Supplementary Fig. 31).

## Discussion

Structural studies of GPCR-arrestin complexes are hampered by the highly dynamic nature of the interaction[55]. Only a few structures are available, all involving rhodopsin-like receptors. Here, non-canonical amino acids for photo- and chemical crosslinking were genetically incorporated into arr2 in living cells to explore its molecular interactions with a secretin-like receptor without artificial stabilization techniques used in conventional structural methods[56]. By combining systematic crosslinking in a cellular environment with extensive molecular modeling and MD simulations, we obtained unique insights into the complex of arr2 with the PTH1R, a prototypical class B GPCR and a major clinical target. After a thorough screening of the

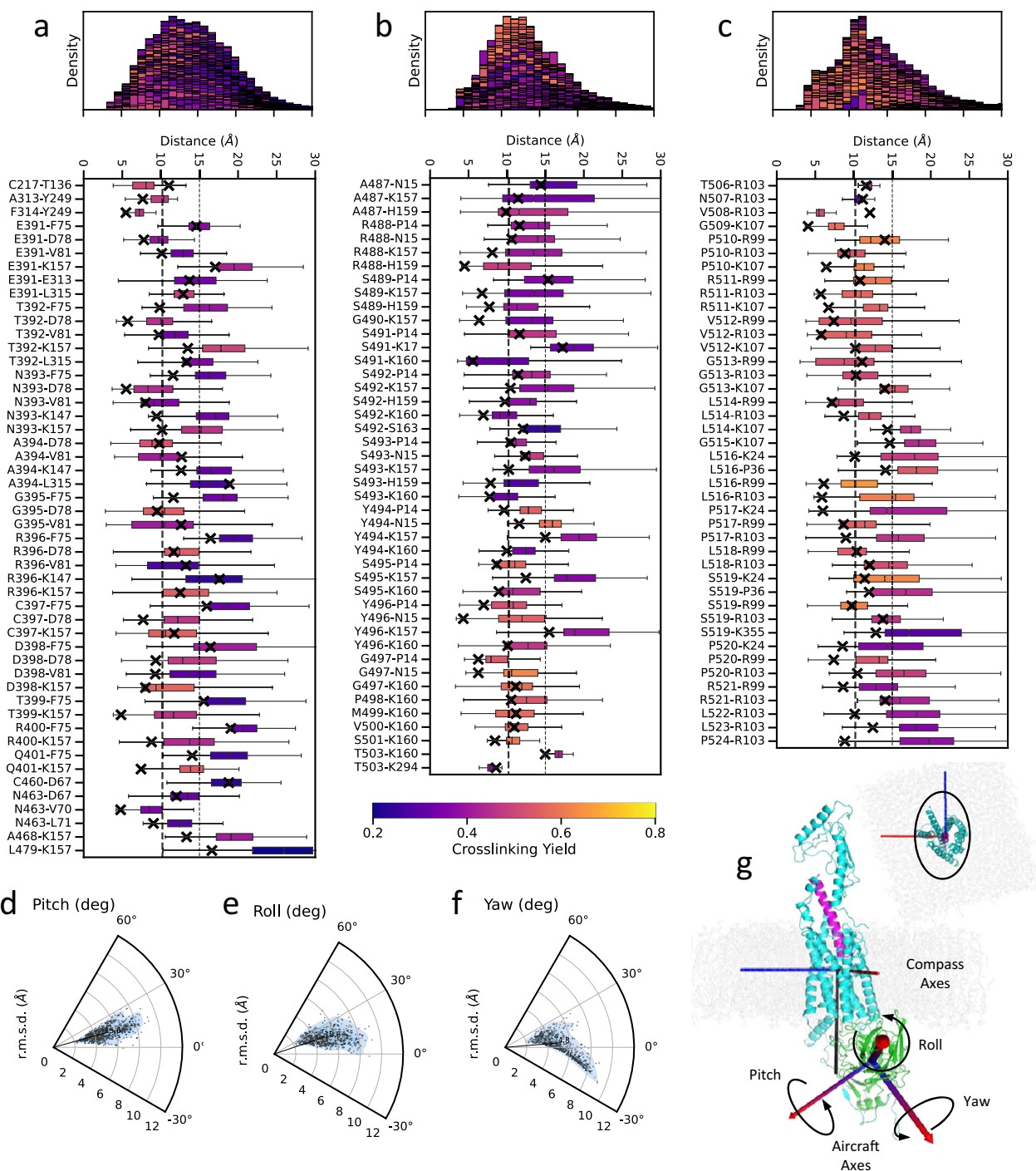

**Fig. 5 | Dynamics of the assembled complex based on M₂R-arr2 template.**
**a**–**c** Box plots and stacked histograms showing statistics of crosslinking distances in MD simulations in PTH1R-arr2. Pairwise Cβ-Cβ distances were measured; when glycine was present, Cα was used instead. Chemical crosslinking pairs are grouped by PTH1R regions, namely (**a**) 7TM + helix VIII (res. D27–F483), (**b**) proximal and distal phosphorylation clusters (res. K484–S504), and (**c**) the rest of the C-terminus in this construct (res. V505–S525). The whiskers of the box plot indicate the maximum and minimum of the distance observed; the center of the box indicates median value of the distance observed; the lower and upper bounds of the box indicates the 25th percentile and the 75th percentile of the distance observed. These descriptive statistics were collected with n = 12,000 frames from the ten 1200 ns trajectories. Distances in the starting model are marked with crosses. The lower bound of the covalent diameter at 10.2 Å of BrEtY is drawn as a thick vertical

dash line accompanied by a thin dash line at 15 Å. Both plots are colored by the crosslinking yield of individual pairs. The stacked histograms on top give an overall summarizing statistics of the Cβ-Cβ distances in each of the PTH1R regions showing that the vast majority of pairs stays within 15 Å distance during MD simulations. The orientation of arr2 relative to PTH1R was measured in terms of pitch, roll, and yaw angles of rotation around their principal axes. Source data for Fig. 5a–c are provided as Supplementary Data 4–6. **d**–**f** Distribution of pitch, roll, yaw angles and r.m.s.d. of arr2 from its starting coordinates for each frame in MD simulations. In general, the orientations are limited by the membrane anchoring of C-edge with pitch angle showing a modest -13 degrees deviation, followed by -17 degrees of roll and up to 25 degrees of yaw deviations. (**g**) Compass and aircraft axes to define pitch, roll, yaw angles. Source data for Fig. 5d–f are provided as Supplementary Data 7.

arr2 surface via a photo-crosslinker, which revealed the footprint of the receptor on the arrestin, proximity-enabled crosslinking at 621 PTH1R-arr2 intermolecular pairs of amino acids provided 136 statistically validated proximity points throughout the large interaction interface. These were translated into soft harmonic restraints to build integrative structural models of the PTHLA-PTH1R-arr2 complex by combining available high-resolution information about its components with cell-derived information on the complex assembly. We had established a similar approach to decipher binding modes of peptide ligands to their class B GPCRs[40,47,48], and have expanded it here to a much more extensive and complex protein-protein interface.

The crosslinking-guided integrative modeling unambiguously determined that the orientation of arrestin in the PTHLA-PTH1R-arr2 complex is similar to that observed in the rhodopsin-arr1, $M_2$R-arr2, and $\beta_1$-AR−arr2 complexes. This supports close similarities between rhodopsin- and secretin-like GPCR interactions with arrestins, while suggesting that the NTS$_1$R-arr2 structures, where arrestin is positioned in a nearly perpendicular orientation, either represent a distinct type of arrestin engagement or reflect specific experimental conditions, as discussed in ref. [21]. In line with all published structures, the segment containing the distal phosphorylation cluster of PTH1R ($^{501}$pSHpTpTVpT$^{506}$) extensively interacts with β-strand I of arrestin in our model, as was observed for the corresponding segments in the C-tail of rhodopsin ($^{340}$pTETpSQV$^{345}$), V$_2$R/V$_2$Rpp ($^{360}$pTApSpSpSL$^{365}$) and NTS$_1$R ($^{409}$pSpSNApTRE$^{415}$). The interaction of the two phosphate groups at pT503/pS504 with K11 and R25 in the arr2 N-terminus breaks the three-element interaction (α-helix I, β-strand I, β-strand XX in the C-terminus) characteristic of basal arrestin conformation[57]. The phosphorylation sites in the distal cluster have been termed "key sites", as they are essential for the formation of a high-affinity arrestin-receptor complex[58]. In PTH1R, the distal cluster was also found to have a greater impact on arrestin recruitment than the proximal cluster[34]. Accordingly, the C-terminally truncated PTH1R variant missing the distal cluster, but not the one missing the proximal cluster, showed impaired arrestin recruitment in our BRET assay (Supplementary Fig. 10).

Our approach revealed additional details of PTH1R-arr2 interactions in regions that were not resolved in available structures. The flexible ICL3 is usually missing in GPCR-arr2 and in most structures of G protein-bound GPCRs, including the PTH1R-Gs complex. Our data clearly show that the ICL3 of PTH1R extensively interacts with arr2, tracing its path from the C-loop contacts (R312, E313, L315) through β-strand VI (D78, V81) to the concave surface of the N-domain (K147) up to the 160-loop (K157). The 160-loop yielded strong crosslinks also with the phosphorylation clusters in the C-tail on the other side. This suggests that the 160-loop of arr2 is dynamically sandwiched between ICL3 and the C-tail of the receptor, which probably contributes to the stabilization of the overall organization of the complex.

Another unresolved region in all existing structures of GPCR-arrestin complexes is located upstream of the key phosphorylation sites. It contains either a phosphorylation cluster, as in the PTH1R (proximal cluster), or negatively charged residues in many GPCRs[58]. Our crosslinking data show that this segment of PTH1R comes close to the positively charged region at the distal edge of the arrestin N-domain ("N-edge"), comprised of the 160-loop and the loop between β-strands IV and V. This arrangement is compatible with the available GPCR-arrestin structures, since the proximal GPCR C-tail is pushed away from the central crest of arrestin by the presence of the 7TM domain, so that this negatively charged region is optimally positioned to interact with the N-edge. This holds true also for the V$_2$R. Although in the structure of arr2 bound to the V$_2$Rpp the proximal phosphorylation lies close to the finger loop, the resolved segment of the receptor C-tail in the full-length V$_2$R-arr2 structure[25] guides the proximal cluster to the arrestin N-edge, in line with our crosslinking results with the full-length V$_2$R (Supplementary Fig. 18c) and with our PTH1R-arr2 model. A similar arrangement with analogous interactions was

observed in the β$_2$V$_2$R-Gs-arr2 megaplex (PDBID: 6ni2)[59] (see Supplementary Fig. 32) and has been suggested by biochemical experiments with rhodopsin long ago[60]. Overall, these observations suggest a function for the N-edge of arrestins in recruiting and/or stabilizing the GPCR C-terminus.

In none of the published structures, the distal C-terminus of a receptor is resolved beyond the key phosphorylation cluster. Our crosslinking data follow the PTH1R C-terminus for at least 10–15 residues further and reveal the existence of an interaction network of positively charged residues in the N-terminus and α-helix I of arr2 (K4, K24, R99) that engage the phosphorylated S519. This pocket might be an accessory recognition feature for phosphorylation sites downstream of the major clusters found in several GPCRs, and may provide additional stabilization of the unlocked three-element interaction in active arrestin[58].

Importantly, crosslinking data do not present a single structural snapshot of the complex, but an average over several conformations. This conformational ensemble likely defines the same functional macro-state. Indeed, while only about half of the pairwise crosslinks strictly satisfy the 10.2 Å Cys-BrEtY Cβ-Cβ distance in the static model, most of them (96%) come within this distance at some point during unbiased MD simulations, with the six outliers coming within 15 Å. The high conformational heterogeneity of the complex is further corroborated by the dynamic variations in the orientation of arr2 relative to the receptor, with a range of motion within 13−25 degrees (Fig. 5d–f, Supplementary Data 7). This is consistent with double electron-electron resonance studies at the rhodopsin-arr1 complex, where variable distances were measured between Y74 in TM2 of rhodopsin (Ballesteros-Weinstein numbering 2.41) and three reference points in arr1, with the most populated distances in all three distributions matching the crystal structure[19]. Variability in the arrestin orientation was also reported for MD simulations of the NTS$_1$R-arr2 and V$_2$R-arr2 complexes[22,25]. While highlighting the dynamics of the PTH1R-arr2 complex, MD simulations revealed its overall stability and robust key interactions within the interface, including both at the 7TM domain and phosphorylation clusters of the C-terminal tail of PTH1R.

In summary, we have provided here an unprecedented insight into structural features of the arr2 complex with the secretin-like PTH1R based on experimental information derived from the physiologically relevant environment of the live cell. Our approach does not require any modifications stabilizing the protein complex and reflects the dynamics of the natural system, allowing us to fill in the gaps in the flexible elements that are missing in known GPCR-arrestin structures. We were able to follow the path of the receptor ICL3 on the arrestin, and unveiled the position of the proximal phosphorylation cluster interacting with a hitherto overlooked positively charged region at the arrestin N-edge. We further reveal the existence of a previously unappreciated interaction network that stabilizes phosphorylation sites downstream of the key sites. As these features are conserved in several GPCRs, our work provides unique complementary information to existing structural data. We believe that this approach will help answer biological questions in the GPCR field that are not addressable with conventional structural methods, while being applicable to protein-protein interactions in general.

## Methods

### Non-canonical amino acids
Bpa was purchased from Bachem. BrEtY was synthesized according to ref. [41] as described below. The ncAAs were stored at −20 °C. Right before the experiments, the needed amount of Bpa was dissolved in 1 N NaOH (aq), whereas BrEtY was dissolved in DMSO.

### Synthesis of O-(2-bromoethyl)-L-tyrosine (BrEtY)
The reaction scheme of the synthesis of O-(2-bromoethyl)-L-tyrosine (BrEtY, **4**) is shown in Supplementary Fig. 33. Briefly, 18.65 g

(60.00 mmol, 1.0 eq.) Boc-L-Tyr-OMe and 26 mL (0.30 mol, 5.0 eq.) of 1,2-dibromoethane were dissolved in 260 mL acetone in a 400 mL reactor system. 25 g (0.18 mol, 3.0 eq.) of potassium carbonate were added and the suspension was refluxed for 21 h at 65 °C. After monitoring, the reaction mixture was cooled down to RT, filtered and the solvent was removed under reduced pressure. The crude product was column purified with cyclohexane/ethyl acetate (2:1) to obtain 9.42 g (23.4 mmol, 39% yield) of the product **2** as a white solid.

In a 500 mL one-neck flask, 19.64 g of **2** (1.0 eq., 48.8 mmol) were dissolved in 100 mL of THF/MeOH (1:1). A solution of 3.51 g of lithium hydroxide (146 mmol, 3.0 eq.) dissolved in 50 mL of $H_2O$ was added and the yellowish solution was stirred for 20 min at RT to complete conversion of the starting material. The reaction was treated with 110 mL 1 N HCl solution (pH of ~2) and the colorless suspension was extracted three times with 100 mL ethyl acetate. The combined organic layers were washed twice with 100 mL brine, dried over $Na_2SO_4$ and filtered. The solvent was removed under reduced pressure to give 18.6 g (48.0 mmol, HPLC purity: 100%) of **3** as a white solid in 98% yield.

In an 1 L one-neck flask, 18.6 g (48.0 mmol, 1.0 eq.) of **3** were dissolved in 100 mL dichloromethane and 47.8 mL (13 eq., 624 mmol) TFA were added. The resulting reaction was stirred for 18 h at RT. The solution was treated with 100 mL methanol and the acidic solvent was removed under reduced pressure. The crude was treated with 150 mL diethyl ether and filtered off to obtain 18.2 g (4.00 mmol) of BrEtY TFA salt (**4**) as a white solid in 90% yield.

$^1$H-NMR: δ [ppm, d$^4$-MeOH, 22 °C] = 7.20 (d, $^3J$ = 8.7 Hz, 2H), 6.92 (d, $^3J$ = 8.7 Hz, 2H), 4.28 (m, 2H), 4.13 (dd, $^3J_{H\text{-}H\text{-}cis}$ = 5.3 Hz, $^3J_{H\text{-}H\text{-}trans}$ = 7.7 Hz, 1H), 3.67 (m, 2H), 3.23 (dd, $^3J_{H\text{-}H\text{-}cis}$ = 5.3 Hz, $^2J$ = 14.6 Hz, 1H), 3.07 (dd, $^3J_{H\text{-}H\text{-}trans}$ = 5.3 Hz, $^2J$ = 14.6 Hz, 1H). $^{13}$C-NMR: δ [ppm, d$^4$-MeOH, 22 °C] = 171.5, 162.7, 159.4, 131.7, 128.2, 120.1, 69.4, 55.5, 36.6, 30.5. ESI-MS: $C_{11}H_{15}Br_1N_1O_3^+$, $M_{calc}$: 288.0230, $M_{found}$ = 288.0239.

## Molecular biology

All enzymes were purchased from New England Biolabs (Ipswich, MA). Cloning was performed in *E. coli* DH5α. For PCR, Phusion High Fidelity Polymerase was used (New England Biolabs GmbH). The ORF of human arr2 was amplified from an RT-PCR sample of HEK293T. Both the ORF of arr2 and PTH1R were cloned into pcDNA3.1 (Thermo Fisher Scientific, Waltham, MA), as described[39]. The sequence for the C-terminal 3xHA affinity tag was obtained by primer extension PCR. A library of arr2-TAG mutants and PTH1R-cysteine or serine mutants was obtained by high-throughput site directed mutagenesis, primers were designed with AAscan[61]. Oligonucleotides were purchased from Microsynth (Balgach, CH) and Biomers (Ulm, DE). All sequences were verified by Sanger sequencing (Microsynth Seqlab Göttingen, DE).

## Cell culture

HEK293T cells were maintained in Dulbecco's modified Eagle medium (DMEM; high glucose 4.5 g/l, 4 mM glutamine, pyruvate; Thermo Fisher Scientific) supplemented with 10% (v/v) fetal calf serum (FCS) (Thermo Fisher Scientific) and 100 U/mL penicillin and 100 µg/mL streptomycin (Thermo Fisher Scientific) (full DMEM) at 37 °C under 5% $CO_2$ and 95% humidity. Cells were passaged at ~80% confluence.

## Photo-crosslinking experiments

HEK293T cells were seeded at 500,000 cells per well in 6-well plates in 2 mL full DMEM. After 24 h the media was exchanged with full DMEM supplemented with 250 µM *p*-benzoyl-L-phenylalanine (Bachem). Cells were transfected using PEI (Polysciences, Warrington, PA) at a PEI:DNA ratio of 3:1 (w/w) in lactate buffered saline (20 mM sodium lactate pH 4 and 150 mM NaCl)[62]. Cells were co-transfected with three plasmids: (1) 900 ng of a plasmid encoding the arr2-stop codon mutant, (2) 900 ng of pIRE4-BpaRS (available from ADDGENE #155342)[39], and (3) 300 ng of a vector encoding PTH1R. 48 h after transfection, the media was

exchanged with 1 mL activation buffer (PBS + 0.1% BSA) supplemented with 200 nM PTH(1-34). After 15 min at 37 °C, the cells were irradiated with UV light in a BLX-365 crosslinker (BioBudget Technologies, Krefeld, DE; 365 nm; 4 × 8 W bulbs) for 15 min on ice. Next, the activation buffer was aspirated and the cells were frozen at −80 °C for 20–30 min, detached with 1 mL PBS supplemented with 1× protease inhibitor cocktail (Roche) and pelleted at 2500 × g for 10 min at 4 °C. Cells were lysed in 80 µL Triton lysis buffer (50 mM HEPES pH 7.5, 150 mM NaCl, 10% glycerol, 1% Triton X-100, 1.5 mM $MgCl_2$, 1 mM EGTA, 1 mM DTT and 1× protease inhibitor) for 30 min on ice and vortexed every 10 min. The samples were centrifuged at 16,000 × g for 10 min at 4 °C to pellet non-soluble debris, supernatants were transferred to pre-chilled tubes. For SDS-PAGE, 4 µL of supernatant was incubated for 30 min at 37 °C in LDS-Sample buffer (250 mM Tris-HCl pH 8.5, 2% (w/v) LDS, 150 mM DTT, 0.4 mM EDTA, 10% (v/v) glycerol and 0.2 mM Coomassie Brilliant Blue G).

## Chemical crosslinking experiments

HEK293T cells were seeded at 500,000 cells per well in 6-well plates in full DMEM. After 24 h, the media was exchanged with full DMEM supplemented with 250 µM BrEtY. Cells were transfected using PEI as described above. Cells were co-transfected with three plasmids: (1) 900 ng of a plasmid encoding arr2-stop codon mutant, (2) 900 ng of XYPylRS/4xM15-tRNA (available from ADDGENE #155343)[39], and (3) 300 ng of a vector encoding indicated PTH1R construct. After 48 h, the media was aspirated and the cells were stimulated for 90 min with 1 mL activation buffer supplemented with 200 nM PTH(1-34). Cell lysis and sample preparation for SDS-PAGE were carried out as described above.

## SDS-PAGE and Western Blot

Samples were resolved on 8% Tris/glycine polyacrylamide gels and transferred to a PVDF membrane (Millipore Immobilon, Merck, pore size 0.45 µm). The membranes were blocked with 5% non-fat dry milk (NFDM) in TBS-T (20 mM Tris-HCl pH 7.4, 150 mM NaCl and 0.1% (v/v) Tween-20) for at least 1 h at RT. Primary antibodies were diluted in 5% NFDM in TBS-T as follows: rat α-HA 3F10 (Roche Diagnostics, Mannheim, Germany) 1:2000; mouse α-PTH1R 4D2 (Thermo Fisher Scientific) 1:2000. Membranes were incubated overnight with the primary antibody at 4 °C under constant gentle agitation, followed by 3 × 10 min washes in TBS-T. Secondary antibodies, either α-rat-HRP for α-HA (Cell Signaling Technology, Danvers, MA) or α-mouse-HRP for α-PTH1R (Santa Cruz Biotechnology, Dallas, TX) were used at dilutions 1:5000 and 1:10,000, respectively, in 5% NFDM in TBS-T. Membranes were incubated for 1 h at RT followed by 3 × 10 min washes in TBS-T. Finally, membranes were soaked in enhanced chemiluminescence reagent (10 parts 0.1 M Tris-HCl pH 8.6 with 250 mg/L luminol, 1 part DMSO with 1100 mg/L *p*-hydroxycoumaric acid and 0.003 parts 30% $H_2O_2$). After 1 min, signals were detected for 5 min in the dark (Gbox, Syngene, Bangalore, IN).

## Densitometric analysis

Raw 16-bit tif images of α-HA blots were imported into western blot detection software (Image Studio Lite, version 5.2, LI-COR, Lincoln, NE). Bands were selected manually using a rectangular tool with a consistent area for the crosslinking band. Crosslinking efficiency was defined as the intensity of the crosslinking band divided by the total intensity of the arrestin signal (bands of crosslinked + not crosslinked arrestin).

## In-cell cAMP accumulation measurement

One day before transfection, ~$2.7 × 10^6$ HEK293T cells were seeded in a 10 cm dish using full DMEM. Cells were transfected with 1 µg of PTH1R-encoding plasmid with or without serine substitutions under control of a CMV promoter, 5 µg of the reporter construct pGL4.29 (humanized PpyRE9 firefly luciferase gene driven by a cAMP-responsive

element and followed by a PEST-sequence (Branchini et al., 2010), 0.5 μg of plasmid pRL encoding Renilla luciferase driven by a CMV promoter and 3.5 μg of the empty pcDNA3.1 vector. Transfection was performed using PEI as described above. The next day, cells were trypsinized and transferred to poly-D-lysine (PDL) coated 96-well plates with a density of ~70,000 cells per well. After 24 h, cells were stimulated at 37 °C for 3 h by the addition of 25 μl of PTH(1–34) dissolved in pure DMEM to final concentrations from $10^{-12}$ to $10^{-6}$ M in a 96-well. Each concentration was analyzed in four wells. The following steps were performed according to Seidel et al. 2017. Cells were washed with ice-cold HDB (12.5 mM HEPES pH 7.4, 140 mM NaCl, 5 mM KCl). Cell lysis was performed using 50 μl of luciferase buffer (10 mM MgSO₄, 25 mM glycylglycine, 4 mM EGTA, pH 7.8) supplemented with 1% Triton X-100 and 1 mM dithiothreitol (DTT) for 30 min on ice under constant gentle agitation. Luciferase activities were measured using a Omega luminometer (BMG LABTECH, Ortenberg, DE) equipped with two injectors. To each well, 50 μl of luciferin substrate buffer (luciferase buffer supplemented with 0.3 mM luciferin, 1 mM ATP, 1 mM DTT pH 7.8) were subsequently added by the first injector and the total luminescence was measured. Afterward, 50 μl of 5 μM colenterazine dissolved in HDB were added to each well (1.67 μM final concentration of coelenterazine in the well). The luminesence of Renilla luciferase was detected using a 475–30 nm emission filter. Firefly luminescence was normalized to the Renilla luminescence. Curves were fitted by non-linear regression using Prism 9 for Windows (Graphpad Software Inc., San Diego, CA). EC₅₀ values were obtained as means with the appropriate CI from at least three independent experiments, each performed in quadruplicate.

## BRET measurements

One day prior to transfection, ~$4.5 \times 10^5$ cells were seeded per well in a 6-well plate. One day later, cells were transfected with 60 ng of the receptor construct PTH1R-NLuc with or without serine substitutions (PTH1R gene coupled to Nanoluc luciferase C-terminally under control of a CMV promoter in pcDNA3.1), 300 ng of construct VE-hArr2 (gene for human arr2 coupled to the Venus gene N-terminally and driven by a CMV promoter in pcDNA3.1) and 1640 ng of empty vector pcDNA3.1 per well. Transfection was performed using PEI as described above. After 24 h, cells were trypsinized and re-seeded to PDL coated 96-wells with a density of ~70,000 cells per well in FluoroBrite™ DMEM (Thermo Fisher Scientific, Waltham, MA). After one day, the medium was replaced by 100 μl of BRET buffer (Gibco™ HBSS (Fisher Scientific, Schwerte, Germany) supplemented with 20 mM HEPES). Afterward, 50 μl of 16.8 μM coelenterazine h dissolved in BRET buffer were added to the wells and incubated for 10 min at 37 °C. Cells were stimulated by the addition of 50 μl of the ligand PTH(1–34) solved in BRET buffer with 0.5% BSA to final concentrations from $10^{-12}$ to $10^{-6}$ M in a well. The fluorescence and luminescence signals were measured 11 min after stimulation using a BMG LABTECH Omega luminometer equipped with 520 nm (for Venus) and 475–30 nm (for NanoLuc) emission filters. The netBRET ratio was calculated by dividing the 520 nm emission by the 480 nm emission. Curves were fitted by non-linear regression using Prism 9 for Windows (Graphpad Software Inc., San Diego, CA). EC₅₀ values were calculated as mean with the appropriate CI from at least three independent experiments, each performed in quadruplicate.

## Constructing optimized static model

Conformational modeling was carried out in ICM-Pro v.3.9.2c (Molsoft LLC). To generate initial rough models of PTHLA-PTH1R-arr2 complexes, all published GPCR-arrestin complexes, including rhodopsin-arr1 (PDBID: 5WOP)[20], β₁-AR–arr2 (PDBID: 6TKO)[24], M₂R-arr2 (PDBID: 6U1N)[23] and NTS₁R-arr2 (PDBID: 6PWC and 6UP7)[21,22], were used as templates to define arrestin conformation and its orientation with respect to the PTH1R receptor. For rhodopsin-arr1, a homology model of arr2 was constructed from the arr1 structure using the sequence

from UniProt ID: Q03431-1 [https://www.uniprot.org/uniprotkb/Q03431/entry#sequences]. An initial PTHLA-PTH1R model was extracted from a cryo-EM model of the PTHLA-PTH1R-Gs complex (PDBID: 6NBF)[33]; missing loops and side-chains were added using loop modeling and optimization tools in ICM-Pro v.3.9.2c. Relative orientations of our PTHLA-PTH1R-arr2 complexes were generated by superposing PTHLA-PTH1R with corresponding structural templates.

To optimize PTHLA-PTH1R-arr2 models in the second stage, crosslinking yields $B_{ij}$ were introduced to the ICM conformational optimization protocol as weights in soft flat-bottomed harmonic potentials $E_{penalty}$ on distance $d_{i\,j}$ between Cβ-Cβ atoms of the corresponding crosslink pair in Eq. (1), where the walls of the flat bottom are set at 4.0 Å and at 10.2 Å.

$$E_{penalty} = \sqrt{B_{ij}} \left[ \max\left(d_{ij} - 10.2,\, 0\right)^2 + \max\left(4.0 - d_{ij},\, 0\right)^2 \right] \quad (1)$$

The lower bound 4.0 Å is selected to penalize atomic clash, while the upper bound 10.2 Å is the maximal Cβ-Cβ distance in the BrEtY-Cys adduct. The entire model then underwent alternating rounds of energy-based optimization (Metropolis Monte Carlo and gradient descent) with and without harmonic restraints using Biased Probability Monte Carlo approach[63] until convergence for a maximum of two million steps. Several regions were exhaustively sampled in the optimization protocol. In the receptor, this includes the stretch L377–F417 in the TM5/ICL3/TM6 region involved in activation, the ICL2 (S308–Y320), helix VIII and the proximal C-term (F461–T525). In the arr2, the N-terminus M1–R7, C-terminus P354–K358 and crest region (R51–L108, P121–V171, N280–S320) were considered flexible. All the other torsion variables were optimized with fast gradient descent. The final refined model is truncated at G530 after our last significant pairwise crosslink at P524; the N-terminus M1–D27 are also truncated as we cannot model these parts with certainty in the absence of further experimental characterizations.

## Molecular dynamics simulations

The M₂R-arr2–based model was uploaded to the Charmm-GUI webserver[64] to generate input files for molecular dynamics (MD) simulations. The structure was embedded into a bilayer of lipids (POPC: Cholesterol in ratio 70:30); initial membrane coordinates were assigned by the PPM server[65] via the Charmm-GUI interface. The molecular content of the system is listed in Supplementary Table 2. All MD simulations were conducted with Gromacs (v.2020.1) simulation engine[66] under Charmm36 force field parameters and topologies[67]. After initial energy minimizations, all systems were equilibrated for 20 ns, followed by production runs of 1200 ns under NVT ensemble with the Nosé-Hoover thermostat. The simulations were performed with GPU clusters at the CARC of the University of Southern California. The total time of trajectories for each model was 12 μs. MD trajectories were analyzed using MdTraj package[68]. In Fig. 5d–g, we defined relative orientations between arr2 and PTH1R using pitch, roll, and yaw angles commonly used to define aircraft rotations[69] (see Fig. 5g for illustration). To monitor these angles along the MD trajectory, principal components (PCs) of arr2 coordinates (Cα only) were calculated on the fly and sorted in descent according to their eigenvalues; the longitudinal axis, vertical axis, and transverse axis of arr2 are defined as PC1, PC2, and PC3, respectively. The direction of the longitudinal axis is fixed by defining nose residues on the arrestin (res. K232–I233, V325–K326, A344–V345); vertical axis always point away from the membrane. Similarly, the principal components of the receptor were used to define the north, east and down axis, where the longitudinal of the receptor is fixed to nose residues on helix IV (res. A333–V336), the vertical points to the intracellular side of membrane normal to the lipid bilayer. The pitch, roll, yaw angles are the angles to align the principal axes of the arrestin $\mathbf{X}_{Arr}$ to that of the

receptor $X_{PTH1R}$. For each frame in a trajectory, we can then solve $X_{PTH1R} = RX_{Arr}$ for the rotation matrix $R$. The directions of these right-handed axes are fixed as described above. Then the pitch $\beta$ roll $\gamma$ yaw $\alpha$ angles can be calculated by Eqs. (2), (3), (4), respectively.

$$\beta = : \arctan 2\left(-R_{31}, \sqrt{R_{11}^2 + R_{21}^2}\right) \tag{2}$$

$$\gamma = : \arctan 2\left(\frac{R_{32}}{\cos(\beta)}, \frac{R_{33}}{\cos(\beta)}\right) \tag{3}$$

$$\alpha = : \arctan 2\left(\frac{R_{21}}{\cos(\beta)}, \frac{R_{11}}{\cos(\beta)}\right) \tag{4}$$

## Reporting summary

Further information on research design is available in the Nature Portfolio Reporting Summary linked to this article.

## Data availability

The authors declare that the data supporting the findings of this study are available within the paper and its supplementary information files. The coordinates of the best PTH1R-arr2 model, based on the $M_2R$-arr2 template, generated in this study have been deposited in the ModelArchive database under accession code ma-2b2xn, other models are deposited under ma-33nf3, ma-5ui3z, ma-9mi1q, ma-f1hkg, ma-v0m33. The following protein structures were used in this paper, accessed via the PDB: 4JQI, 6NBF, 5W0P, 6U1N, 6TKO, 6PWC, 6UP7, 1G4M, 7R0C, 6NI2, 6NI3. Source data are provided with this paper.

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

## Acknowledgements

This work was supported by the German Research Foundation (DFG Grants: CO822/3-1 and CO822/3-2 to IC; CRC 1423, project number 421152132, subproject B04 to IC), by the National Institute of Diabetes and Digestive and Kidney Diseases grant P01 DK011794 (VK), and the National Institute of General Medical Sciences (NIGMS) R35 GM122491 (VVG). We thank the staff at the USC Center for Advanced Research Computing for providing computational resources and Felix Nitzsche for the synthesis of BrEtY in Leipzig.

## Author contributions

I.C. conceived the project and supervised the crosslinking part. T.B. and Y.A. designed and performed all crosslinking experiments, and interpreted results. Y.A. performed the densitometric analysis of the data. S.E. performed the arrestin recruitment experiments with truncated PTH1R. A.F. performed the functional assays at PTH1R-mutants. M.T. designed and supervised the large-scale synthesis of BrEtY. V.K. conceived and supervised the computational part of the project. J.H.L. performed statistical analysis of the data, all modeling and MD experiments, and interpreted results. B.Z. performed preliminary modeling

experiments. The manuscript was written by Y.A., T.B., and J.H.L. and was revised by V.K., V.V.G., and I.C.

## Funding

## Competing interests

The authors declare no competing interests.
