## [Peer Review File · Nature Communications]

Structural details of a Class B GPCR - arrestin complex revealed by genetically encoded crosslinkers in living cellsREVIEWER COMMENTS

Reviewer #1 (Remarks to the Author):

The paper developed a model for the interface of the PTH1R-arrestin complex, based on constraints derived from a cleverly designed exhausting set of crosslinking experiments followed by Monte Carlo simulation. The model thus developed was studied by molecular dynamics simulation to confirm its validity.

I wonder if the authors checked whether the cross-linked complexes are active. My guess is that such a check could identify cross links that resulted in non-physiological complexes and thus could be excluded from the constraint list.

Also, there are a few minor issues whose clarification would improve the manuscript, listed below.

Summary: This is nit picking but to me 'dynamic stability' sounds like an oxymoron. I know what you mean but still ...

L98: Maybe a more explicit term could be used instead of 'Intermolecular proximity'?

L108-109: What is 'permissive position' (I have a guess, but it is better to be explicit)?

L140-141: Is 'PTH1R footprint' the same as 'permissive position'? If yes, try to keep the notation consistent. If not, explain the difference.

L145: Help the reader by telling if the 'six positions' are on arr2 or on PTH1R.

Reviewer #2 (Remarks to the Author):

Aydin et al. provide an exhaustive analysis of the interaction landscape of arrestin2 (arr2) with the parathyroid hormone 1 receptor 1 (PTH1R) in cells using genetically encoded cross linkers. One reason I use the term "exhaustive" is because one does get exhausted teasing apart all the figures and begins to wonder why Nature Comm does not pay their reviewers for the hours of effort especially while they are on vacation. Humor aside, this is very rigorous work. Key to the analysis is the substitution of the genetically encoded cross linkers at nearly every position in arrestin2. While one would never expect that all of these substitutions would yield functional protein (even if they express well), just by mass action one would still expect to generate enough meaningful data for the computational analysis that one has a very good chance at getting at the right answer. It is also quite rigorous because it went after these complexes with several different cross linking approaches.

I am enthusiastic about the work for several reasons. First of all, they characterize (admittedly at low resolution) the interaction of arrestin2 with a GPCR. This is not yet present in the literature. Second, it is perhaps the first of an arrestin with a Class B receptor (although not surprisingly it is not in the end too much different from that of Class A complexes). Third, this data gets at the dynamical nature of the complex in a way that crystallography and cryo-EM cannot. Cryo-EM has the best chance to get at dynamics because it can, in principle, reveal multiple configurations, but typically only the most prominent are revealed (and even these may not represent a large fraction of the possible states). The work also suggests several novel interactions that may be features of other arrestin complexes, where residues in the receptor are conserved.

Those are the pros. The biggest con is that this technique probably cannot discriminate models in which arrestin is bound by tail alone, versus TM core alone, versus both. So even though we get a dynamic picture of a bimodal interaction, it still may not be reflective of the true conformational range of the complex in cells because one only gets data for when residues come close to each other for enough time for cross linking to take place. The most robust interaction in the end seems to involve the tail of the receptor with arrestin2, and this seems to indicate to me that a large fraction of the complexes might be tail only, but given that cross-linking efficiency is based on many things, it may be impossible to discriminate. A second con is that given the nature and effort behind this technique, it seems unlikely to be widely applied to many other gpcr arrestin complexes unless one had significant time and resources...and would one in the end get enough discriminatory differences between different data sets for one to ever say there will be a functional difference given the resolution of the technique? A third con is that given that this is photo cross linking over 15 minutes of exposure (or longer with chemical cross linking), it samples a broad range of possible dynamics, and it is impossible to say which crosslinks reflect the functionally relevant complexes in the cell. A fourth is that the most reactive groups are often the most dynamic (lowest affinity). So there is a bias in the analysis here, potentially, towards data reflecting the least important interactions. The authors get at this point with their "hubs" diagram, but it is unclear to me how the presence of these highly reactive elements bias the simulations. Have they

tried simulations where they "silence" these regions and compare the results with simulations with all restraints?

Major editorial comments:

1) Please address the 4 cons, at least in the discussion. For example, I think it would be good to have a discussion of possible modes of arr2 binding based on this sort of analysis. Are we, as implied by the result, only expecting tail+core (bimodal) interactions? If tail-only is also a possibility, then should we be concerned as much with pitch/roll/yaw? Is the core interaction really important, or it is the tail, as implied by the writeup? Is the MD analysis sufficient in length and free enough of assumptions to discriminate between unimodal (tail only) or bimodal modes? What happens if they start with a tail mode?

2) Authors, unfortunately, should include insights from the new vasopressin-arrestin1 cryo-EM structure (bioRxiv, 2022.2002.2011.480047) in their analysis instead of the vasopressin receptor tail peptide crystal structure, which is artifactual for reasons the authors have explained. In fact I don't consider the crystal structure useful here for other reasons, including the use of Fab30. This structure is not formally published just yet, but I bet the authors would share coords or conjectures could be made from the figures therein at least.

3) The movement of H8 is pretty dramatic compared to prior structures. It might be right. But based on Figure 3 it seems to be driven by relatively few crosslinks. In particular with A468. Can we be confident in this result? Or maybe it just gets disordered. Is its interactions with the membrane perturbed in the simulations? Looks like it is pulled away, which seems unreasonable if true.

Minor editorial comments:

1) Please do not capitalize "western".

2) Page 3. In introduction, probably should note that there is a vasopressin-arr1 complex structure in addition to the others.

3) Page 19. "Currently, no structural information is available for any GPCR beyond the key phosphorylation sites." Needs to be revised for clarity because taken by itself, this statement obviously isn't true.

4) Fig. 1. directional turn arrows are ambiguous in direction (and also imply 360° rotations).

- 4b) Fig. 2. The chemical cross linking data seems to indicate that the distal tail sites in the tail are not important. I understand why this might be, but this highlights I think the "fourth con" above.
- 5) Fig. 3. I have to say I don't know what to make of panel A and related figures in the supplement. It is too detailed for a mere human to pick apart and unless the authors can explain the take-aways better, I'd consign it to the supplement.
- 6) Fig. 5. Given my arguments about tail vs. bimodal modes of interaction, is this sort of analysis misleading? What is the reader supposed to take away from the histograms at the top and why are they multicolored?
- 7) Fig. S1. Caption on my computer has a box instead of a beta.
- 8) Fig. S5. Please be consistent and reasonable with use of significant figures. Probably only 2 (e.g. "40." instead of "39.9") for both EC50 means and confidence intervals (e.g. "3-30" for wt).
- 9) Fig. S7. The shading and colors used make reading some of the values difficult.
- 10) Fig. S9. See point 8 above. Many of the values have 4 significant figures. 1-2 seem warranted.
- 11) Fig. S10. I couldn't understand panels C and D of this figure and text labels are quite small in C. Maybe it is for the MD audience.
- 12) Fig. S11-16. Needed? Or please explain what the reader should be looking for in comparing these.
- 13) Fig. S17. Comment. Does the more closed conformation of the cytoplasmic cleft imply that this could be a way in which ligands can impose bias in class B receptors (unclear to me if this receptor has known bias). Or is this loop just too dynamic anyhow? As above, I have qualms about the modeling of H8 in panel B. Plus if it is pulled away from the membrane (and perhaps the TM core), there is a good chance it is disordered (not a helix anymore).
- 14) Fig. S18. Still not getting what I am supposed to be tracking in panel A. Is it the downward right motion of the very-hard-to-read labeled data points? Although the orange data points are shifting, the overall distribution of data points seem very similar. What to make of that then?
- 15) Fig. S20. Please label the figure and the various hubs or exclude the other colored hubs.
- 16) Fig. S26. Text hard to read
- 17) Table S1. I very much appreciated this analysis. Just a comment. Does the literature therefore support the notion that GRK2, which only seems to phosphorylate the proximal site, is not as important for arr binding as the GRK/other kinase that labels the distal site? Is the GRK4 subfamily then implied to be the key regulators of PTH1R desensitization?

The extensive nature of my comments is not meant to convey a lack of enthusiasm. There is just a lot to process. I think it is nice work. Regards, John Tesmer

Reviewer #3 (Remarks to the Author):

The manuscript by Coin and colleagues describes an impressive tour-de-force study on the interaction of a GPCR with beta-arrestin using genetically encoded crosslinker amino acids to obtain distance constraints. The study first identifies positions on arrestin that when substituted with the photo-crosslinker pBPA crosslink to the GPCR. Next, the suitable sites are used to install a proximity-reactive amino acid to produce crosslinks to native or genetically introduced cysteines on the GPCR, which allows to identify distance constraints. Subsequently, the distance constraints are used in a modelling study to improve structural information of the complex.

This studies sets new standards in the way unnatural amino acid crosslinking is used to investigate protein-protein interactions. The insights obtained on the GPCR/arrestin complex add significantly to the understanding of the structure and function of this important protein family.

The quality of the data is very high and well controlled. The presentation is of high quality and suitable for publication in its present form.

To make this story even more compelling I would recommend an experiment that builds on the new structural model. This could be further crosslinking or mutagenesis experiments or some other functional study.

Reviewer #4 (Remarks to the Author):

The authors present a model of GPCR (PTHR) – arrestin complex derived from cross-links. First, they incorporated a broad-specificity cross-linking unnatural amino acid in every position of arrestin, and determined the interaction footprint of the PTHR on arrestin. Subsequently, they used more specific cross-linking unnatural amino acid that interacts only with cysteines. This allowed them to obtain more selective constraints to computationally build a very detailed model of the complex, revealing the overall geometry of the complex similar to 5 out of 6 presently available GPCR-arrestin structures. However, the real novelty of their approach to provide structural information about transient interaction – something that has been “invisible” so far, explaining what happens to the ICL3 of the receptor, as well its very long C terminus. This work is likely to stimulate further studies of the molecular architecture of GPCR signalling.

The manuscript is very well written, and the conclusions are well justified. I can only congratulate the authors on producing such a spectacular work.

Point-by-point responses.

Reviewer #1 (Remarks to the Author):

The paper developed a model for the interface of the PTH1R-arrestin complex, based on constraints derived from a cleverly designed exhausting set of crosslinking experiments followed by Monte Carlo simulation. The model thus developed was studied by molecular dynamics simulation to confirm its validity.

I wonder if the authors checked whether the cross-linked complexes are active. My guess is that such a check could identify cross links that resulted in non-physiological complexes and thus could be excluded from the constraint list.

We thank the reviewer for the interesting suggestion. Testing functional properties of the crosslinked complexes, however, would be very challenging since (1) a varying ratio of crosslinked and non-crosslinked complexes exist in cells, crosslinking yields are never quantitative (2) both mutations themselves and crosslinks can modulate function dynamically even if the complex has overall "physiological" conformation.

At the same time, we should note that our analysis of proximity crosslinking distinguishes between specific and non-specific binding (background). In other words, our crosslinking system does not capture significant amounts of complexes arising from random collision, but only complexes that remain associated in a suitable orientation and long enough for the chemical reaction to happen. This ensures that the captured here crosslinks are relevant to native-like interactions between PTH1R and arrestin.

Also, there are a few minor issues whose clarification would improve the manuscript, listed below.

Summary: This is nit picking but to me 'dynamic stability' sounds like an oxymoron. I know what you mean but still ...

We agree that this expression is inadequate. We have changed it to "stability and dynamic nature".

L98: Maybe a more explicit term could be used instead of 'Intermolecular proximity'?

Thank you for pointing this out. We have changed "points of intermolecular proximity" with "intermolecular pairs of proximal amino acids", which is the exact definition of what we determine experimentally. We need to keep the concept of "proximity" and cannot exchange it with "contact" or "interaction", because crosslinking methods in general can only reveal enriched/stabilized spatial closeness of the involved moieties, but neither direct contact nor interaction of the residues that are replaced with the crosslinkers.

L108-109: What is 'permissive position' (I have a guess, but it is better to be explicit)?

Thank you for drawing our attention to this. In this sentence the term "permissive" is indeed misleading. We rephrased in a more precise way: "Bpa was well tolerated, with 94% of the mutants yielding full-length protein." We meant that permissive positions allow successful readthrough of the stop codon, yielding full-length target protein with the Bpa modification.

L140-141: Is 'PTH1R footprint' the same as 'permissive position'? If yes, try to keep the notation consistent. If not, explain the difference.

Thank you for spotting this confusing element. The concepts of "permissive" and "PTH1R footprint" are indeed different. For "permissive" see comment right above. The "PTH1R footprint" refers to the

set of arrestin positions that crosslink the receptor. In other words, the “PTH1R footprint” is the photocrosslinking pattern mapped on the arr surface, shown in Fig. 1 and Supplemental Fig. 4.

We have added a short sentence at the end of the second paragraph on page 5, where we define the term “footprint”: “The set of arrestin positions photo-crosslinking with PTH1R represents the footprint of the receptor on the arrestin”

L145: Help the reader by telling if the ‘six positions’ are on arr2 or on PTH1R.

Thank you! Edited as suggested, now: Arr2 with BrEtY at six positions located in the finger loop (T58, E66, F75, D78), the middle loop (T136) and the 160-loop (E155) yielded the strongest signals.

Reviewer #2 (Remarks to the Author):

Aydin et al. provide an exhaustive analysis of the interaction landscape of arrestin2 (arr2) with the parathyroid hormone 1 receptor 1 (PTH1R) in cells using genetically encoded cross linkers. One reason I use the term "exhaustive" is because one does get exhausted teasing apart all the figures and begins to wonder why Nature Comm does not pay their reviewers for the hours of effort especially while they are on vacation. Humor aside, this is very rigorous work. Key to the analysis is the substitution of the genetically encoded cross linkers at nearly every position in arrestin2. While one would never expect that all of these substitutions would yield functional protein (even if they express well), just by mass action one would still expect to generate enough meaningful data for the computational analysis that one has a very good chance at getting at the right answer. It is also quite rigorous because it went after these complexes with several different cross linking approaches.

I am enthusiastic about the work for several reasons. First of all, they characterize (admittedly at low resolution) the interaction of arrestin2 with a GPCR. This is not yet present in the literature. Second, it is perhaps the first of an arrestin with a Class B receptor (although not surprisingly it is not in the end too much different from that of Class A complexes). Third, this data gets at the dynamical nature of the complex in a way that crystallography and cryo-EM cannot. Cryo-EM has the best chance to get at dynamics because it can, in principle, reveal multiple configurations, but typically only the most prominent are revealed (and even these may not represent a large fraction of the possible states). The work also suggests several novel interactions that may be features of other arrestin complexes, where residues in the receptor are conserved.

We thank the reviewer for his positive and motivating comments!

Those are the pros. The biggest con is that this technique probably cannot discriminate models in which arrestin is bound by tail alone, versus TM core alone, versus both. So even though we get a dynamic picture of a bimodal interaction, it still may not be reflective of the true conformational range of the complex in cells because one only gets data for when residues come close to each other for enough time for cross linking to take place. The most robust interaction in the end seems to involve the tail of the receptor with arrestin2, and this seems to indicate to me that a large fraction of the complexes might be tail only, but given that cross-linking efficiency is based on many things, it may be impossible to discriminate.

Please see the answer to the tail/core question below, at point 1 of "major editorial comments", which addresses in detail this very issue.

A second con is that given the nature and effort behind this technique, it seems unlikely to be widely applied to many other gpcr arrestin complexes unless one had significant time and resources...

We totally understand the concern of the reviewer. Indeed, the Coin lab is afraid that no more PhD students will want to join the group after this work is published. To address this problem, we have already established a crosslinking assay in 96-well format that allows circumventing the western blot analysis, which is the current bottleneck of the whole procedure. As we intend to present this assay in a separate publication, we do not feel we should stress this aspect in this manuscript. In fact, we believe that our approach could become a standard complementary analysis for low-resolution structures, for addressing regions of protein complexes that are too flexible to be resolved in X-ray of Cryo-EM investigations, and/or for validation of structures in more physiological conditions of living cells.

and would one in the end get enough discriminatory differences between different data sets for one to ever say there will be a functional difference given the resolution of the technique?

We anticipate that the technique allows distinguishing features of a GPCR-arrestin complex formed with the same receptor phosphorylated by different GRKs (this is our new ongoing study).

A third con is that given that this is photo cross linking over 15 minutes of exposure (or longer with chemical cross linking), it samples a broad range of possible dynamics, and it is impossible to say which crosslinks reflect the functionally relevant complexes in the cell.

We absolutely understand the concern of the reviewer. However, we should note that our crosslinking system, and especially chemical crosslinking, distinguishes between specific and non-specific binding (background). In other words, our crosslinking system does not capture significant amounts of complexes arising from random collision, but only complexes that remain associated in a suitable orientation and long enough for the chemical reaction to happen. Although the proteins can dynamically assume a broad range of conformations over the crosslinking time, the crosslinks belong to functionally relevant macro-states. We have addressed this issue in the last paragraph of the discussion: "Importantly, crosslinking data do not present a single structural snapshot of the complex, but an average over several conformations". We have added "**This conformational ensemble likely defines the same functional macro-state.**" Indeed, almost all amino acid pairs involved in crosslinks come within the crosslinking distance in MD simulations, representing only modest changes in the conformation of the complex. It is in principle possible that we are capturing distinct macro-states, but we do not see compelling evidence for this in our data, which all converge to the same arrangement.

Though photo-crosslinking results were not used in the final pairwise crosslink selection and model building, we do plan to improve the temporal resolution of our experiments by applying more powerful lamps and diazirine photo-crosslinkers that allow for shorter irradiation time. In contrast, shortening the time of chemical crosslinking may be a very challenging chemical endeavor, although some photo-activatable ncAAs for chemical crosslinking have recently been reported.

A fourth is that the most reactive groups are often the most dynamic (lowest affinity). So there is a bias in the analysis here, potentially, towards data reflecting the least important interactions. The authors get at this point with their "hubs" diagram, but it is unclear to me how the presence of these highly reactive elements bias the simulations. Have they tried simulations where they "silence" these regions and compare the results with simulations with all restraints?

We hope we interpret the comment of the reviewer correctly in saying that "the most reactive groups" implies the residues of arrestin that give the largest number of crosslinks, i.e. the crosslinking hubs. The chemical reactivity in our experiments is otherwise always the same (same BrEtY-crosslinker bait and same Cys-prey). While it is true that some hubs e.g. K157 on the arrestin 160-loop reacted with 20 different PTH1R residues, the crosslinking pairs show a range of different crosslinking intensities (see plots below). As crosslinking hubs usually comprise a mixture of strong and weak hits, we do not expect crosslinking hubs to introduce significant bias while building the model.

With respect to MD simulations, all our simulations are totally free of any restraints. The fact that regions including hubs do not fall apart during simulation, but – on the contrary - almost all amino acid pairs involved in hubs come within the distance of crosslinking confirms that these regions are indeed hubs of proximity in the complex.

Additional analysis: We have prepared a few plots to better illustrate the crosslinking state of the hubs (see image below). First, we plotted the relative intensity of crosslinks against the number of PTH1R crosslinking partners per Arr2 residue (left), and against the number of arr2 crosslinking partners per PTH1R residue. In both cases, we found no significant correlation between the number of crosslinks and crosslinking intensity.

Second, we plotted the RMSF in MD per arr2 residue against the number of PTH1R crosslinking partners per Arr2 (left panel below), finding again no significant correlation. The same conclusion holds for PTH1R (right panel) residues.

Major editorial comments:

1) Please address the 4 cons, at least in the discussion.

We have addressed points 2,3 and 4 above. Point 1 follows.

For example, I think it would be good to have a discussion of possible modes of arr2 binding based on this sort of analysis. Are we, as implied by the result, only expecting tail+core (bimodal) interactions? If tail-only is also a possibility, then should we be concerned as much with pitch/roll/yaw? Is the core interaction really important, or it is the tail, as implied by the writeup?

The reviewer addresses here a very important point, i.e. the existence of a “tail” and a “tail + core” conformation of GPCR-arrestin complexes, according to literature reports.

We have performed experiments to specifically check whether our method allows distinguishing the two conformations. It is generally accepted that the deletion of the finger loop of arrestin (Δ FLR-arr2)

leads to the assembly of GPCR-arr complexes in the “tail-only” conformation, where arrestin lacks the interaction with the receptor core. We have performed crosslinking experiments with Δ FLR-arr2 carrying Bpa in the N-domain and we did not observe crosslinking signals above background. This result suggests that our crosslinking system does not capture the tail-only conformations of the PTHR1-arr2 complex. We have included these results in the manuscript, added a paragraph in the results section, and a new figure in the SI (see below).

In the manuscript:

To investigate whether our crosslinking pairs may derive from two distinct populations of complexes either in the “tail” or in the fully engaged “tail + core” conformation, we have incorporated the photo-crosslinker Bpa in the N-domain (N-terminus, 160-loop) of an arrestin variant depleted of the finger loop, which does not interact with the receptor core⁴⁴ (Supplementary Fig. 8b). Finger loop-depleted Bpa-arr2 were not able to capture the PTH1R, whereas the corresponding full-length arr2 variants carrying the crosslinker at the same positions yielded strong crosslinking signals in western blot. These results suggest that the crosslinking constraints reflect the predominance of the fully engaged conformation

In the SI:

Supplementary Fig. 9. Crosslinking Experiments with arr2 variants lacking the finger loop.

Bpa was incorporated at the indicated positions in an arr2 variant where residues Y63 to K77 (finger loop) were deleted (Δ FLR-arr2). Bpa was incorporated at positions that gave chemical crosslinking with the C-tail of the PTH1R (supplementary Fig. 6 and 7) and are distributed on the N-terminus and the 160-loop of arr2. Cells co-expressing wt PTH1R and either Δ FLR-arr2 or full-length arr2 were treated with PTH(1-34) and irradiated with UV light (365 nm). We show the western blots analysis of the cell lysates stained with an α -HA antibody. The numbering of arrestin positions is identical to wt.

Although it has been reported in the literature that arrestin with deleted finger-loop can form stable complexes with the β 2AR-V2R chimera in a tail-only conformation (10.1073/pnas.1701529114), this observation has been done in a reconstituted system in vitro where the tail conformation was stabilized by the antigen-binding fragment 30 (Fab30). Latest single-molecule FRET experiments failed to capture arr2 tail-only engagement of the phosphorylated receptor in absence of Fab30 stabilization (Asher et al. Cell 2022, doi:10.1016/j.cell.2022.03.042). In general, most structural and in vitro studies of GPCR-arr complexes had to use chimeras with the V₂Rpp to ensure complex stability, and V₂Rpp carries a lot more phosphates than most WT receptors.

Overall, our data suggest that what we capture with our crosslinking is predominantly the core+tail fully engaged conformation.

Is the MD analysis sufficient in length and free enough of assumptions to discriminate between unimodal (tail only) or bimodal modes? What happens if they start with a tail mode?

Thank you for the suggestion. The 1.2 μ sec MD simulations in this study were designed to assess stability and dynamic flexibility of the modeled complex, but were not expected to capture large-scale rearrangements, like the rearrangement occurring between “tail only” and “bimodal” interactions. Likewise, the limited scale MD simulations starting from “tail only” mode are not expected to spontaneously converge to fully engaged mode, which is likely to need more than 1000 times longer MD runs. To directly test the reviewer's suggestion, we developed several tail-only models as starting points for the complex simulations by applying the subset of crosslinks at the C-tail of PTH1R residues (downstream of position 487, both phosphorylation clusters are included), which is supposed to represent the tail-only interaction. In contrast to the fully engaged complex, where models converged to the same orientation, the tail-only modelling yields at least 4 very distinct models, not engaging 7TM. Three of these models have the finger loop completely disengaged from the 7TM, whereas the fourth model has the finger loop around the hinge of TM7-H8. Some of these have N-lobe at the intracellular cleft of 7TM.

We run 5*1.2 microseconds of MD simulation on each of these 4 models, and did not find any of the runs with arrestin to spontaneously couple with 7TM core in the limited time of simulation. As expected, the arrestin and the receptor sample a wide range of orientations, while staying apart from each other (see Supplementary Figure 31d). Besides, even when some of these tail-only trajectories (4 out of 20 trajectories) spontaneously come closer to each other, the finger loop never gets fully engaged with the 7TM. The divergence of these tail-alone from fully engaged MD simulations suggests that tail-only mode of interaction cannot explain the crosslinking data for finger loop and other parts of arr2 engaged with 7TM of the receptor.

As discussed above, our new crosslinking experiments show that Δ FL-arr2, which cannot engage 7TM, does not crosslinks to C-terminal residues of the receptor suggesting that “C-tail only” binding mode is weak and transient, at least at the experimental conditions we used. Therefore, we include below the following figure to report the results of these C-tail only simulations, but we do not feel that these data should be included in the manuscript.

Legend: Intercouple Distance in MD simulations started tail-only modelling. The tail-alone models are illustrated on the right-hand-side. Time series of intercouple distance, defined as the distance between mean of arrestin (green sphere) and mean of 7TM core res.182-460 (cyan sphere) (see Supplementary figure 31d). A white sphere is shown to illustrate the location of arrestin in the fully engaged model.

2) Authors, unfortunately, should include insights from the new vasopressin-arrestin1 cryo-EM structure (bioRxiv, 2022.2002.2011.480047) in their analysis instead of the vasopressin receptor tail peptide crystal structure, which is artifactual for reasons the authors have explained. In fact I don't consider the crystal structure useful here for other reasons, including the use of Fab30. This structure is not formally published just yet, but I bet the authors would share coords or conjectures could be made from the figures therein at least.

We have updated the manuscript in light of the structures that were published during the revision time. First, we have mentioned the solved V2R-arr2 and 5HT2B-arr2 complex structures and referenced them in the introduction.

Recently, the complexes of the full length V₂R^{REF} and the serotonin 5-HT_{2B} receptor^{REF} with arr2 have been published.

Second: In the results section, we have re-written the part describing the modeling and the crosslinking validation of the N-edge path in a more compact way, which reduces the emphasis on the phosphopeptide structure. We still cite that structure, though, as it is still the only one where the proximal cluster is resolved:

We validated this position of the proximal phosphorylation cluster at the external edge of the N-domain by assessing whether an alternative pose involving interactions with the finger loop, as it was observed in the structure of the arr2-V₂Rpp complex (Supplementary Figure 18a), is reconcilable with our crosslinking data. First, when crosslinking arr2 with full-length V₂R, we did find strong crosslinking hits in the 160-loop, but not in the finger loop (Supplementary Fig. 18b). Second, when adding distance restraints between the proximal cluster in the PTH1R C-tail and strand VI of arr2 in a modeling experiment (Supplementary Fig. 19 a-b), the level of strain in this region of the complex dramatically increased compared to our model (Supplementary Fig. 19c). Overall, these data confirm the path of the proximal phosphorylation cluster of PTH1R C-tail at the N-edge of arr2.

The corresponding Supplementary Fig. 18 has been modified and includes a view of the Cryo-EM structure (it is be called in the discussion).

Third: In the discussion, we have rewritten the part about the N-edge and we focus now on the hints from full-length structures, including the V2R-arr2 structure. The phosphopeptide structure is only mentioned.

This arrangement is compatible with the available GPCR-arr structures, since the proximal GPCR C-tail is pushed away from the central crest of arrestin by the presence of the 7TM domain, so that this negatively charged region is optimally positioned to interact with the N-edge. This holds true also for the V₂R: Although in the structure of arr2 bound to the V₂Rpp peptide the proximal phosphorylation lies close to the finger loop, the resolved segment of the receptor C-tail in the full-length V₂R-arr2 structure²⁵ guides the proximal cluster to the arrestin N-edge, in line with our crosslinking results with the full-length V₂R (Supplementary Fig. 18c) and with our PTH1R-arr2 model. A similar arrangement with analogous interactions was observed in the β₂V₂R-Gs-arr2 megaplex (PDBID: 6ni2)⁶⁰ (see Supplementary Fig. 32) and has been suggested by biochemical experiments with rhodopsin long ago⁶¹. Overall, these observations suggest a function for the N-edge of arrestins in recruiting and/or stabilizing the GPCR C-terminus.

3) The movement of H8 is pretty dramatic compared to prior structures. It might be right. But based on Figure 3 it seems to be driven by relatively few crosslinks. In particular with A468. Can we be

confident in this result? Or maybe it just gets disordered. Is its interactions with the membrane perturbed in the simulations? Looks like it is pulled away, which seems unreasonable if true.

We thank the reviewer for raising this question and we agree that the non-canonical conformation of H8 in this model may need further explanation. First of all, apart from A468 and L479 crosslinking with K157 in 160-loop of arr2, several residues proximal to H8 (A468, R467 and S489) also crosslinking with 160-loop and residues 14-16 of arrestin are not compatible with canonical H8 position. Residue K472 also makes a potentially strong crosslink to K157 (see SI Fig.7), but was deemed non-significant due to the large error margin. While we agree that all these crosslinks could potentially be modeled in a highly permissive case of disordered H8, the pattern of crosslinks every 3 or 4 residues (468-472-479) suggests α -helical secondary structure in H8.

The MD simulations also support helical conformation of H8 between G464 and A480, as we have shown in a new SI figure Fig. 28), although we understand that this does not constitute proof of such a conformation. We also assessed the orientation of H8 regarding the helical bundle by measuring the corresponding angles. The angle between TM7b H8 shows 124.3° on average with a SD of 9.1° suggesting that H8 is oriented at a slight angle compatible with interaction with the membrane. We have added these data in SI Fig 29). As we mentioned above, however, we cannot exclude a possibility that crosslinking data in H8 region can be explained by alternative flexible loop conformation(s), and we now mention this possibility in the text:

This non-canonical conformation is supported by crosslinking of Helix VIII residues of PTH1R A468, L479 and proximal positions A468, R467, S489 with K157 and H159 of arr2, by ionic interactions within the receptor helix VIII (Fig 4e) involving K471^{PTH1R}-pS489^{PTH1R}, R485^{PTH1R}-pS489^{PTH1R}/pS493^{PTH1R}, as well as by stability of Helix VIII in MD simulations (see below). However, our crosslinking dataset cannot exclude an alternative situation, where residues of helix VIII lack secondary structure and are flexible.

Minor editorial comments:

1) Please do not capitalize "western".

We have edited the manuscript accordingly.

2) Page 3. In introduction, probably should note that there is a vasopressin-arr1 complex structure in addition to the others.

We have mentioned in the introduction the work of Bous et al. and also the new structure of the arrestin complex with the serotonin receptor as mentioned above.

3) Page 19. "Currently, no structural information is available for any GPCR beyond the key phosphorylation sites." Needs to be revised for clarity because taken by itself, this statement obviously isn't true.

Thank you for pointing this out. We have rephrased: In none of the published structures, the distal C-terminus of a receptor is resolved beyond the key phosphorylation cluster.

4) Fig. 1. directional turn arrows are ambiguous in direction (and also imply 360° rotations).

We have improved the visualization in Fig. 1 and in SI Fig. 1. We hope it is clearer now.

4b) Fig. 2. The chemical cross linking data seems to indicate that the distal tail sites in the tail are not important. I understand why this might be, but this highlights I think the "fourth con" above.

The reviewer touches here an intrinsic paradox of all types of crosslinking experiments, i.e. that crosslinking can map points close to a crucial interaction, but cannot precisely spot the interaction itself, as the interacting partners are substituted by the crosslinkers. In our case, we do have only relatively few pairwise hits directly within the distal cluster. However, the regions in the immediate vicinity are very well sampled, which allows to model this region with confidence.

Our BRET assay also indicates that the distal cluster is an important element in the complex. We have added the following sentence to the discussion: **Accordingly, the C-terminally truncated PTH1R variant missing the distal cluster, but not the one missing the proximal cluster, showed impaired arrestin recruitment in our BRET assay (SI Fig. 10).**

5) Fig. 3. I have to say I don't know what to make of panel A and related figures in the supplement. It is too detailed for a mere human to pick apart and unless the authors can explain the take-aways better, I'd consign it to the supplement.

We understand that Fig 3a plot is rather dense in information. However, this type of plot (distance between the residues in crosslinking pairs, as measured in the model, plotted against their respective crosslinking yield) is essential, as we feel it is the most straightforward way to describe and evaluate a crosslinking-guided model. We revised the caption of the figure to incorporate the take-aways more clearly, and added the following explanation:

The horizontal dashed line at 10.2 Å marks the estimated radius ($C\beta$ - $C\beta$) for BrEtY-Cys crosslinking, whereas the dotted line at 15.0 Å represents the maximal crosslinking distance when taking into account the flexibility of the complex. The vast majority of the crosslinking pairs (125 out of 136), lie within 15.0 Å in the static 3D model, suggesting its overall agreement with the crosslinking data.

6) Fig. 5. Given my arguments about tail vs. bimodal modes of interaction, is this sort of analysis misleading? What is the reader supposed to take away from the histograms at the top and why are they multicolored?

As described above, we are confident that our experimental constraints and the corresponding models in Figures 3 and 4 describe the fully engaged tail+core conformation, as we have now explicitly written in the result section.

These results suggest that the crosslinking constraints reflect with large predominance the fully engaged conformation

The MD restraint-free analysis of model represents dynamic variations in the fully engaged complex, which is overall stable and maintains local distances between proximal pairs. The histograms in Figure 5 serve as a summary of all individual histograms below them. They are built by stacking individual distance histograms for crosslink pairs (each rectangle in the histogram column refers to an individual crosslink pair in the box plot). They are colored according to the crosslinking yield (see the color bar above panel f). The key takeaway from these histograms is that in the MD simulation without any distance restraints the crosslinking pairs still concentrate around the lower bound of the distance of BrEtY crosslink as a macroscopic observable of the ensemble average. Another important observation is that their long tails are dominated by crosslinks with weak yields (dark blue)

We have added following explanation in the legend:

Both plots were colored by the crosslinking yield of individual pairs. The stacked histograms on top give an overall summarizing statistics of the C β -C β distances in each of the PTH1R regions showing the vast majority of pairs staying within 15 Å distance during MD simulations.

7) Fig. S1. Caption on my computer has a box instead of a beta.

We have replaced the sign and we hope that the issue is fixed now.

8) Fig. S5. Please be consistent and reasonable with use of significant figures. Probably only 2 (e.g. "40." instead of "39.9") for both EC50 means and confidence intervals (e.g. "3-30" for wt).

Thank you for this remark. We have fixed the numeric values in the table.

9) Fig. S7. The shading and colors used make reading some of the values difficult.

We improved the legibility of the values by enlarging and recoloring the matrix.

10) Fig. S9. See point 8 above. Many of the values have 4 significant figures. 1-2 seem warranted.

Thank you for this remark. We have fixed the numeric values in the table.

11) Fig. S10. I couldn't understand panels C and D of this figure and text labels are quite small in C. Maybe it is for the MD audience.

We thank the reviewer for this helpful comment. Please note that this figure has become now Figure S11. We have simplified the figure and removed panel C, because the important information is already included in the other panels. The labels are now enlarged, and we have expanded the caption to include more detailed explanation of the whole figure.

12) Fig. S11-16. Needed? Or please explain what the reader should be looking for in comparing these.

Thank you for this comment. As we have mentioned in point 5, we consider these plots reporting the intermolecular distances against experimental crosslinking yield as the best way of analyzing a crosslinking-guided model. We have reported the plot corresponding to the best model in the main text (Fig. 3A) and we would like to keep the plots of the discarded model in the SI for the reader who might be interested in comparing details of the results obtained with different templates.

13) Fig. S17. Comment. Does the more closed conformation of the cytoplasmic cleft imply that this could be a way in which ligands can impose bias in class B receptors (unclear to me if this receptor has known bias). Or is this loop just too dynamic anyhow?

We would like to thank the reviewer for this suggestion. Indeed, PTH1R in our model features a slightly more closed conformation of the cytoplasmic cleft compared to the G protein-bound conformation in the cryo-EM structure (PDB 6nbf). This observation agrees with other arr2-bound experimental structures, where the TM5/6 are drawn slightly more inward to the cleft with respect to G protein bound structures. This conformational preference, indeed, might serve as a basis for ligand selectivity between the two pathways.

We have added this comment to the legend of Fig S17:

The inward movement of TM5/6 in our model is in line with the observation in other arrestin-bound experimental structures that the TM5/6 are slightly closer (drawn inward to the cleft) compared to the respective G-protein bound complexes.

14) Fig. S18. Still not getting what I am supposed to be tracking in panel A. Is it the downward right motion of the very-hard-to-read labeled data points? Although the orange data points are shifting, the overall distribution of data points seem very similar. What to make of that then?

We thank the reviewer for this helpful comment. We have completely rewritten the legend of this figure and we hope that the message is clear now. Please note that we have rearranged the SI and this figure has become now Figure S19.

15) Fig. S20. Please label the figure and the various hubs or exclude the other colored hubs.

We thank the reviewer for this suggestion. The figure is now labeled, and we have drawn 3 examples of hubs at different regions of arr2 for clarity.

16) Fig. S26. Text hard to read

We thank the reviewer for pointing to this. The labels in the figure are now enlarged.

17) Table S1. I very much appreciated this analysis. Just a comment. Does the literature therefore support the notion that GRK2, which only seems to phosphorylate the proximal site, is not as important for arr binding as the GRK/other kinase that labels the distal site? Is the GRK4 subfamily then implied to be the key regulators of PTH1R desensitization?

We thank the reviewer for this interesting question. Taken together the literature does not allow the conclusion that the GRK4 subfamily is the key driver PTH1R desensitization or that GRK2 only targets the proximal site. From the older literature (Blind et al. 1996 and Malecz et al 1998) the role of GRK2 targeting the distal cluster remains unclear, while other studies did not investigate GRK subtypes. To avoid confusion, we have removed the “GRK2-mediated” from the remark to the study of Malecz et al. 1998 in SI Table 1, since the authors identified the phosphorylation sites belonging to the proximal cluster in HEK293 without controlling which subtypes are present.

A recent study by Haider et al. (Nat. Comm. 2022) shows that both GRK2 and GRK4 subfamilies can target both the proximal and distal phospho-cluster. In a cellular setting in which only one of the four GRKs 2,3,5 or 6 was present, the authors tested arr2 recruitment to PTH1R mutants containing just one of the two phosphorylation clusters using BRET assays. They found that for both clusters, overexpression of GRKs belonging to both subfamilies could increase arrestin recruitment.

The extensive nature of my comments is not meant to convey a lack of enthusiasm. There is just a lot to process. I think it is nice work. Regards, John Tesmer

Reviewer #3 (Remarks to the Author):

The manuscript by Coin and colleagues describes an impressive tour-de-force study on the interaction of a GPCR with beta-arrestin using genetically encoded crosslinker amino acids to obtain distance constraints. The study first identifies positions on arrestin that when substituted with the photo-crosslinker pBPA crosslink to the GPCR. Next, the suitable sites are used to install a proximity-reactive amino acid to produce crosslinks to native or genetically introduced cysteines on the GPCR, which allows to identify distance constraints. Subsequently, the distance constraints are used in a modelling study to improve structural information of the complex.

This studies sets new standards in the way unnatural amino acid crosslinking is used to investigate protein-protein interactions. The insights obtained on the GPCR/arrestin complex add significantly to the understanding of the structure and function of this important protein family.

The quality of the data is very high and well controlled. The presentation is of high quality and suitable for publication in its present form.

To make this story even more compelling I would recommend an experiment that builds on the new structural model. This could be further crosslinking or mutagenesis experiments or some other functional study.

We thank the reviewer for their positive comments and for their suggestion. Indeed, we have planned a series of functional experiments as a follow-up study that will include the dissection of the phosphorylation pattern at the PTH1R and the contributions of different GRKs.

To make this set of experiments meaningful and convincing, a large number of receptor and arrestin mutants need to be tested in different experimental settings. We feel that this extensive work belongs to a separate project and goes beyond the scope of this manuscript.

Reviewer #4 (Remarks to the Author):

The authors present a model of GPCR (PTHR) – arrestin complex derived from cross-links. First, they incorporated a broad-specificity cross-linking unnatural amino acid in every position of arrestin, and determined the interaction footprint of the PTHR on arrestin. Subsequently, they used more specific cross-linking unnatural amino acid that interacts only with cysteines. This allowed them to obtain more selective constraints to computationally build a very detailed model of the complex, revealing the overall geometry of the complex similar to 5 out of 6 presently available GPCR-arrestin structures. However, the real novelty of their approach to provide structural information about transient interaction – something that has been “invisible” so far, explaining what happens to the ICL3 of the receptor, as well its very long C terminus. This work is likely to stimulate further studies of the molecular architecture of GPCR signalling.

The manuscript is very well written, and the conclusions are well justified. I can only congratulate the authors on producing such a spectacular work.

We thank the reviewer for this positive and encouraging evaluation.

REVIEWERS' COMMENTS

Reviewer #1 (Remarks to the Author):

The authors fully addressed all the issues I raised.

Reviewer #2 (Remarks to the Author):

The authors have done a very thorough job of addressing my major criticisms and I remain enthusiastic about the work.

The one thing I would ask them to reconsider/modulate is their interpretation of their delta-finger loop control experiment that fails to drive cross linking. There is data out there now, including but not limited to my lab, that suggests the finger loop could be mediating contacts with membrane (or detergent as the case may be). Thus another interpretation of the loss of function would be that loss of the finger loop reduces affinity not because of the loss of a core interaction, but because of loss of interactions with the membrane surrounding the receptor.

If they agree that that may be a possibility, the edit I would suggest is that they just state that as an alternative (but perhaps unlikely?....they can decide) possibility. Seems the bulk of the data supports a core complex forming. I like the control experiment overall.

Reviewer #3 (Remarks to the Author):

I congratulate the authors on their impressive study and recommend publication in the present form.

Kind regards,

Heinz Neumann

Reviewer #4 (Remarks to the Author):

After reading the comments of other referees and the authors' responses, I feel that the authors improved the manuscript significantly and addressed the comments of other referees very well. I did not have any reservations about this work in the original version.

Point-by-point responses.

REVIEWERS' COMMENTS

Reviewer #1 (Remarks to the Author):

The authors fully addressed all the issues I raised.

Reviewer #2 (Remarks to the Author):

The authors have done a very thorough job of addressing my major criticisms and I remain enthusiastic about the work.

The one thing I would ask them to reconsider/modulate is their interpretation of their delta-finger loop control experiment that fails to drive cross linking. There is data out there now, including but not limited to my lab, that suggests the finger loop could be mediating contacts with membrane (or detergent as the case may be). Thus another interpretation of the loss of function would be that loss of the finger loop reduces affinity not because of the loss of a core interaction, but because of loss of interactions with the membrane surrounding the receptor.

If they agree that that may be a possibility, the edit I would suggest is that they just state that as an alternative (but perhaps unlikely?...they can decide) possibility. Seems the bulk of the data supports a core complex forming. I like the control experiment overall.

Reviewer #3 (Remarks to the Author):

I congratulate the authors on their impressive study and recommend publication in the present form.
Kind regards,
Heinz Neumann

Reviewer #4 (Remarks to the Author):

After reading the comments of other referees and the authors' responses, I feel that the authors improved the manuscript significantly and addressed the comments of other referees very well. I did not have any reservations about this work in the original version.

We are grateful to all reviewers for their constructive and encouraging comments and we hope that the manuscript now satisfies your requirements.

Best regards

Irene